# Process-Based Modeling of the High Flow of a Semi-Mountain River under Current and Future Climatic Conditions: A Case Study of the Iya River (Eastern Siberia)

Andrey Kalugin 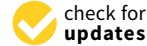

Water Problems Institute, Russian Academy of Sciences, 119333 Moscow, Russia; andrey.kalugin@iwp.ru

**Abstract:** The purpose of the study was to analyze the formation conditions of catastrophic floods in the Iya River basin over the observation period, as well as a long-term forecast of the impacts of future climate change on the characteristics of the high flow in the 21st century. The semi-distributed process-based Ecological Model for Applied Geophysics (ECOMAG) was applied to the Iya River basin. Successful model testing results were obtained for daily discharge, annual peak discharge, and discharges exceeding the critical water level threshold over the multiyear period of 1970–2019. Modeling of the high flow of the Iya River was carried out according to a Kling–Gupta efficiency (KGE) of 0.91, a percent bias (PBIAS) of −1%, and a ratio of the root mean square error to the standard deviation of measured data (RSR) of 0.41. The preflood coefficient of water-saturated soil and the runoff coefficient of flood-forming precipitation in the Iya River basin were calculated in 1980, 1984, 2006, and 2019. Possible changes in the characteristics of high flow over summers in the 21st century were calculated using the atmosphere–ocean general circulation model (AOGCM) and the Hadley Centre Global Environment Model version 2-Earth System (HadGEM2-ES) as the boundary conditions in the runoff generation model. Anomalies in values were estimated for the middle and end of the current century relative to the observed runoff over the period 1990–2019. According to various Representative Concentration Pathways (RCP-scenarios) of the future climate in the Iya River basin, there will be less change in the annual peak discharge or precipitation and more change in the hazardous flow and its duration, exceeding the critical water level threshold, at which residential buildings are flooded.

**Keywords:** runoff generation; process-based modeling; floods; the Iya River; the town of Tulun; change in maximum flow; the ECOMAG model; ISIMIP

## 1. Introduction

The impact of climate change is expressed not only in the variability of annual and seasonal river runoff, but also in an increase in the frequency of extreme hydrological events, the main share of which is due to floods and low water levels. At the current rate of global warming, the risk of flooding formation increases significantly. According to [1], the intensity and duration of floods for the period since the last decade of the 20th century have been increasing globally while, since 2004, changes are accelerating, and the frequency of severe floods is unprecedented. Areas vulnerable to flooding will increase globally, depending on the intensity of future warming [2]. The overall risk of global floods will increase by 187% compared to the scenario without climate change [3]. According to the AOGCMs ensemble of the Coupled Model Intercomparison Project (CMIP), the range of growth in agricultural land area and population affected by a twofold increase in the frequency of floods, respectively, varies by 7 and 15 times, depending on the choice of model by the middle of the 21st century (e.g., [4]).

An overview of future climate change based on climate projections indicates an overall increase in extreme precipitation, which is globally consistent with current trends [5]. An increase in the intensity of rainfall contributes to an increase in the frequency of catastrophic

floods in Russia [6,7]. This process is especially typical for the territories of the Far East, southern Siberia, and the European part of Russia [8]. According to the data of station observations in the territory of Russia, intense rainfall and floods are more often observed in the mountainous regions of the Caucasus, Urals, Altai, Eastern Sayan, Trans-Baikal and the south of the Far East [9,10]. Since local land surface conditions and previous extreme precipitation are more influential, it is difficult to determine the change in frequency and severity of floods associated with anthropogenic impacts of global warming [11].

Since the beginning of the 21st century, several catastrophic floods of rain origin have occurred in Russian territory. Among such extreme events, local floods in mountainous or coastal river basins can be distinguished: in the entire Kuban River basin in 2002, the Adagum River in 2012, rivers of the Black Sea coast in the Krasnodar Territory in 2010, 2015, and 2018, and in the Primorye Territory in 2015 and 2016. These floods were caused by extreme precipitation for several days [9,12]. As a result of most of these floods, the maximum discharge over the observation period was exceeded. However, for example, a large-scale flood occurred over the entire Amur River basin in the period of July–September 2013, which became a national disaster. It is shown in [13,14] that this flood was the result of an extremely rare combination of unfavorable hydrometeorological factors: the formation of a high-altitude frontal zone, along which deep, moisture-saturated cyclones continuously moved for two months, and high pressure blocking over the Pacific Northwest, which prevented the movement of these cyclones from the continent towards the Sea of Okhotsk. The result of such synoptic macroprocesses was the formation of extremes, both in terms of volume and duration of rainfall, on a scale of almost the entire basin. In some parts of the Amur basin, precipitation from July to August exceeded the annual norm.

This study is about the analysis of floods in the Iya River basin located in the Irkutsk Region (Eastern Siberia). Over the past 40 years, several catastrophic floods have occurred in the Iya basin which have caused significant socioeconomic damage to the Tulun district [15]. The last and highest flooding over the observed period was in 2019. At the end of 2019, the results of adaptation of the two-dimensional hydrodynamic model RiverFlow 2D near the town of Tulun were presented [16]. due to the lack of hydrometeorological measurements and difficulty of using simple flow forecasting methods such as the flow transformation along the channel or various statistical relationships, a detailed analysis of the generation of floods on the Iya River and future changes in high flow should be carried out by a physically based modeling method. Accordingly, the development of a process-based hydrological model with distributed parameters (the main method of this study) using a continuous long-term period of hydrometeorological observations from 1970 to 2019, including recent years, and its verification by the accuracy of calculating high runoff, will make it possible to carry out a comprehensive analysis of the formation conditions of the highest floods in a unified methodological approach. This is one aspect of the purpose of this study. Another aspect is that the applied method of process-based hydrological modeling makes it possible to obtain the results of the influence of future climate changes on the physical transformation of the high flow of the Iya River, which may differ significantly from the historical period. A similar analysis of the influence of runoff factors on the 2013 flood in the Amur River basin was carried out based on the results of spatially distributed runoff generation modeling [17]. It is recommended that the methods of physically-based hydrological modeling, together with the AOGCMs, be applied to assess the impact of future climate changes on the formation of high rainfall runoff on semi-mountain rivers (e.g., [18–21]). Although it can be argued that physically based hydrologic models could be the best choice for modeling rainfall–runoff processes in data-scarce watersheds, the need for comprehensive comparisons of the performance of various lumped conceptual model structures is strongly felt in regions with limited data availability (e.g., [22,23]).

## 2. Materials and Methods

### 2.1. Study Basin and Semi-Distributed Hydrological Model

Analysis of publications in the abstract and citation databases Web of Science and Scopus showed that there were no physically based runoff generation models for the Iya River basin at the time of the 2019 flood. In this regard, a model based on ECOMAG software [24] was created before the gauge near the town of Tulun, taking into account the main processes of runoff generation, including infiltration of precipitation, snow melting, freezing and thawing of the soil, evapotranspiration, prechannel, subsurface and groundwater flow, and streamflow transformation in the channel system.

In a model schematization of a river basin, its surface was divided into subbasins (hydrological response units–HRUs) based on a digital elevation model (DEM) and the structure of the river network. Modeling of hydrological processes at any HRU was performed for four levels: the topsoil layer, horizon of caliche, groundwater and prechannel flow. During the cold season, snow cover was added. Prechannel flow was formed after filling the depressions of the land surface due to the formation of excess water, which did not infiltrate and flowed down the slopes of the catchment to the river network. As a result of its infiltration into the soil, water moved in the aeration zone along a slope to the channel network or transformed into a groundwater zone. In the model, the subsurface and groundwater flow was described according to the Darcy equation, and the prechannel and stream flow was described by the kinematic wave equation. The total porosity in the soil aeration zone was divided into capillary and non-capillary zones. It was assumed that, when soil moisture was less than the field capacity, all soil water was in the capillary zone, and when soil moisture was greater than the field moisture capacity, it was in the non-capillary zone. In conditions of high soil moisture, the actual evaporation was equal to the potential, and then it decreased linearly to zero as the soil moisture decreased to wilting point. Potential evaporation was estimated according to the Dalton method.

The snowmelt rate was calculated using the degree–day method. The phase transformation of precipitation depended on the air temperature. The evaporation of solid and liquid phases of snow was estimated using data on the air humidity deficit. It was assumed that vertical temperature profiles in snow and frozen and thawed soil differ insignificantly from linear ones, and moisture migration to the soil freezing front was insignificant. Under these conditions, the dynamics of the depth of freezing and thawing of the soil could be described by a system of differential equations. The infiltration of rain and melt water into frozen soil was calculated taking into account the effect of ice content in frozen soil on the hydraulic conductivity of the soil. A more detailed mathematical description of the flow generation processes in the ECOMAG model is presented by its author in [25,26].

The source of the Iya River is in the northeast of the Eastern Sayan, and the mouth is in the Bratsk reservoir (the Yenisei River basin). The total catchment area is 18,100 km$^2$, up to the town of Tulun is 14,500 km$^2$. The water regime of the Iya River is characterizedby low spring floods, high summer floods, and winter runoff due to groundwater. June–August are the most abundant months, accounting for almost 60% of the annual river runoff. There are two functioning gauges of Roshydromet on the Iya River: Arshan, 192 km from the source, and Tulun, 365 km from the source (119 km to the Bratsk reservoir). The mean annual discharge of the Iya River at the Arshan and Tulun gauges for the observation period was 88 m$^3$ s$^{-1}$ and 150 m$^3$ s$^{-1}$, respectively. The Iya River belongs to the class of medium-sized rivers (catchment area of 2000 to 50,000 km$^2$). However, the Iya River basin has well-defined mountainous and lowland areas (Figure 1). The upper part of the Iya River basin, up to the Arshan gauge, belongs to the mountain type of rivers, taking into account the large values of the longitudinal slope (about 0.38%) and, accordingly, the streamflow velocity, as well as the rocky bottom. In the section of the Iya River between the Arshan and Tulun gauges, the longitudinal slope is 0.09%, i.e., the value is similar for lowland rivers. However, if considering the catchment area to the Tulun gauge, then the Iya River can be classified as a semi-mountain river. The mountainous part corresponds to the boundaries of the river basin to the Arshan gauge, with a catchment area of 5140 km$^2$

and an average elevation of 1490 m a.s.l., calculated using the HYDRO1k DEM with a resolution of 1 km. HYDRO1k data are available online (https://www.usgs.gov/centers/eros/science/usgs-eros-archive-digital-elevation-hydro1k accessed on 12 March 2021). Another so-called lowland part of the river basin up to the Tulun gauge is 9360 km², with an average elevation of 710 m a.s.l.

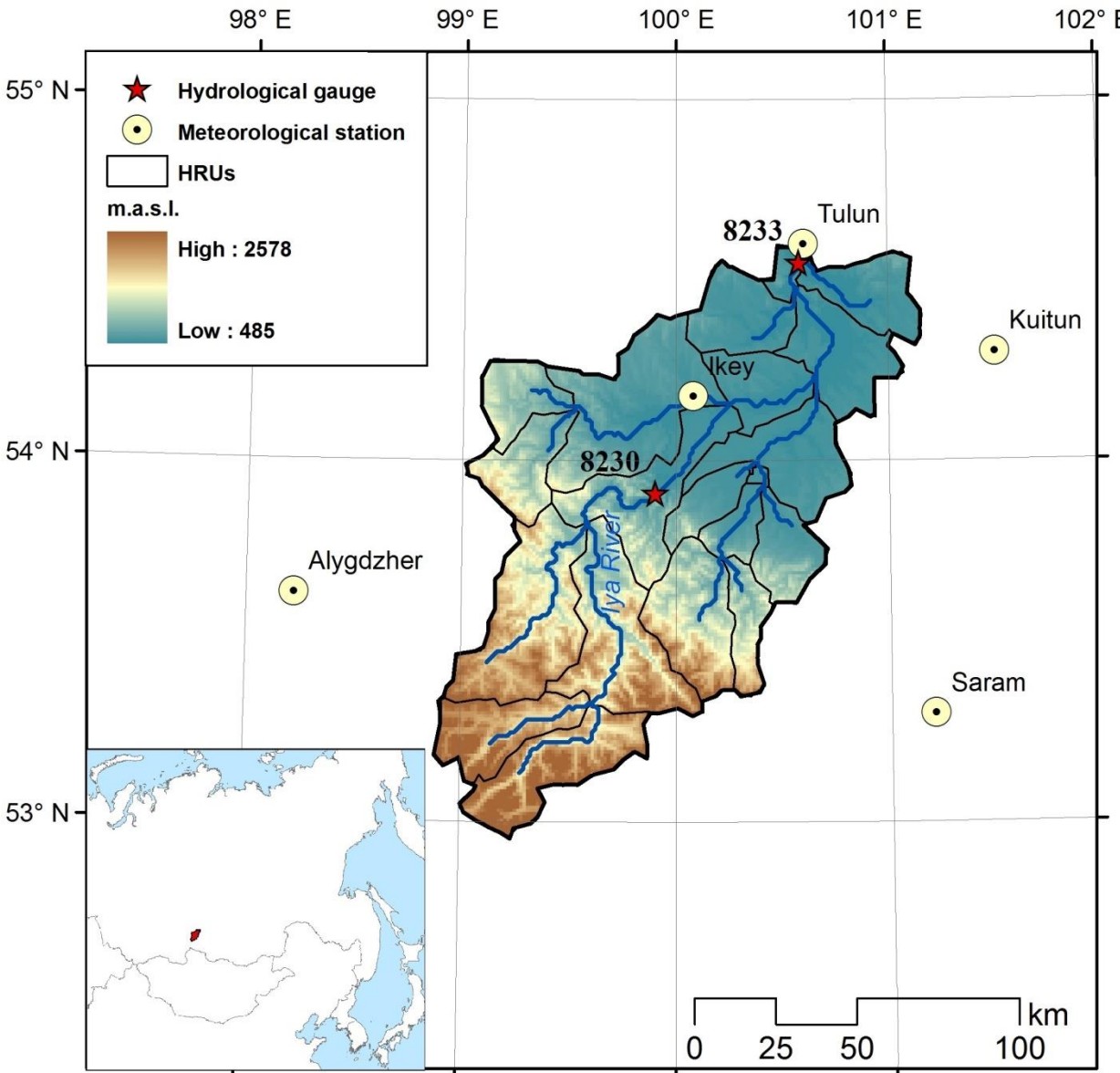

**Figure 1.** Location of hydrometeorological stations and hydrological response units of the runoff generation model in the Iya River basin (8230—Arshan gauge, 8233—Tulun gauge).

There are only two functioning meteorological stations in the Iya River basin: Tulun and Ikey. However, they are located in the lowland part of the river basin. The data from the Saram and Alygdzher meteorological stations, which refer to the basins of the bordering Oka and Uda rivers, respectively, were used to account for changes in the meteorological values of the mountainous area. In addition, data from the Kuitun meteorological station, located 30 km northeast of the Iya River, were used to interpolate the data on the lowland part of the basin. Thus, the boundary conditions of the runoff generation model are the daily precipitation, air temperature and humidity, measured at 5 meteorological stations (Table 1). Taking into account the large height difference between the source and the mouth

of the river, as well as the location of meteorological stations at an altitude of up to 1000 m, it became necessary to determine the precipitation gradient of 40 mm per 100 m of elevation for the mountain part of the catchment, under calibration of the ECOMAG model.

**Table 1.** Information about meteorological stations, daily data of which are used as boundary conditions in the runoff generation model over the period 1970–2019.

| Index of the World Meteorological Organization | Name | Latitude, ° N | Longitude, ° E | Altitude, m a.s.l. |
|---|---|---|---|---|
| 29894 | Alygdzher | 53.63 | 98.22 | 1031 |
| 30504 | Tulun | 54.6 | 100.6 | 487 |
| 30505 | Kuitun | 54.3 | 101.5 | 520 |
| 30507 | Ikey | 54.18 | 100.08 | 510 |
| 30605 | Saram | 53.3 | 101.2 | 622 |

The model was based on global databases of land surface parameters (Figure 2): Harmonized World Soil Database (HWSD–available online accessed on 12 March 2021 http://www.fao.org/soils-portal/data-hub/soil-maps-and-databases/harmonized-world-soil-database-v12) was used for 13 soil types (the top 3 by share of the catchment area: Ferric Podzols 32%, Eutric Podzoluvisols 27%, Gelic Regosols 9%) and Global Land Cover Characterization (GLCC–available online accessed on 12 March 2021 https://www.usgs.gov/centers/eros/science/usgs-eros-archive-land-cover-products-global-land-cover-characterization-glcc) was used for 24 types of landuse/landcover (LULC) (the top 3 by share of the catchment area: cool mixed forest 25%, small leaf mixed woods 22%, low sparse grassland 11%). GLCC resolution is 1 km. Imagery dates from April 1992 through to March 1993.

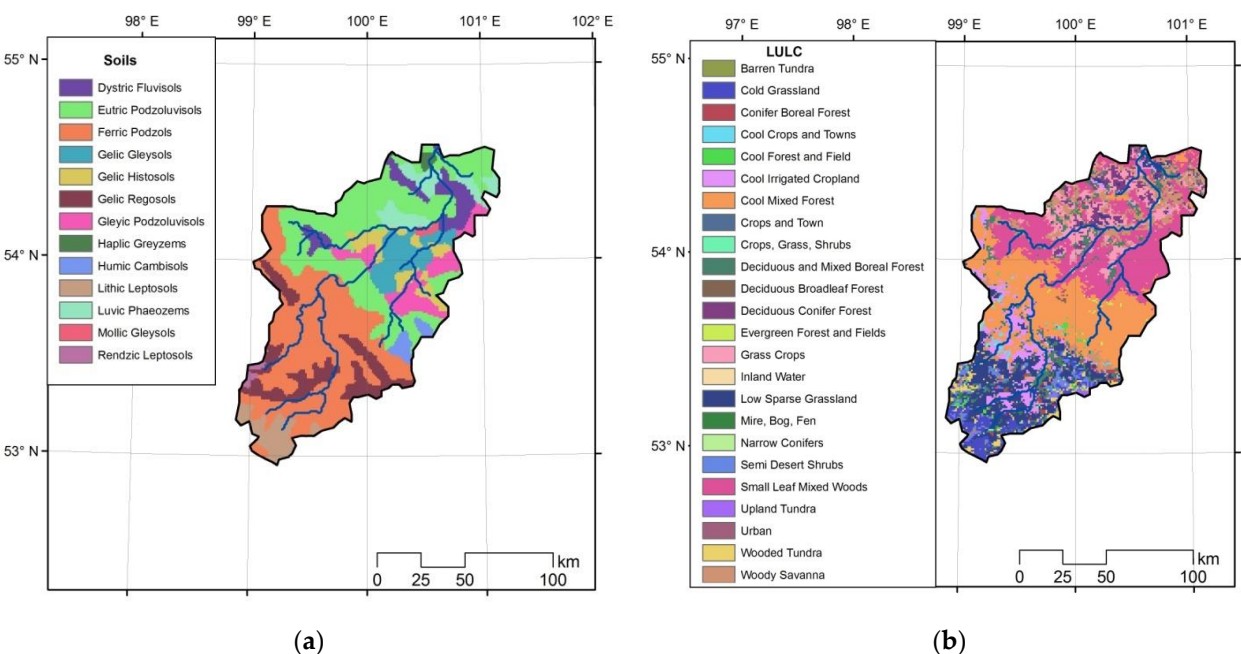

(**a**)  (**b**)

**Figure 2.** Soil map (**a**) and landuse/landcover map (**b**) of the Iya River basin, according to Harmonized World Soil Database and Global Land Cover Characterization.

The average annual air temperature in the Iya River basin is −2.5 °C and the summer temperature is 13.2 °C, over the period 1970–2019 (Table 2). At the same time, the mean annual and summer air temperature in the mountainous part of the basin is lower than in the lowland part by 3.3 °C and 4.3 °C, respectively. The annual precipitation in the

Iya River basin is 562 mm, and summer precipitation is 321 mm. Precipitation in the mountainous part of the catchment is 40% higher than in the lowland part (Figure 3).

**Table 2.** Hydrometeorological characteristics of the Iya River basin over the period 1970–2019.

| Characteristics | The Whole River Basin | Mountainous Area | Lowland Area |
| --- | --- | --- | --- |
| Mean annual temperature, °C | −2.5 | −4.6 | −1.3 |
| Summer temperature, °C | 13.2 | 10.4 | 14.7 |
| Annual precipitation, mm | 562 | 690 | 491 |
| Summer precipitation, mm | 321 | 404 | 276 |
| Annual runoff depth, mm | 327 | 542 | 209 |
| Summer runoff depth, mm | 184 | 325 | 107 |

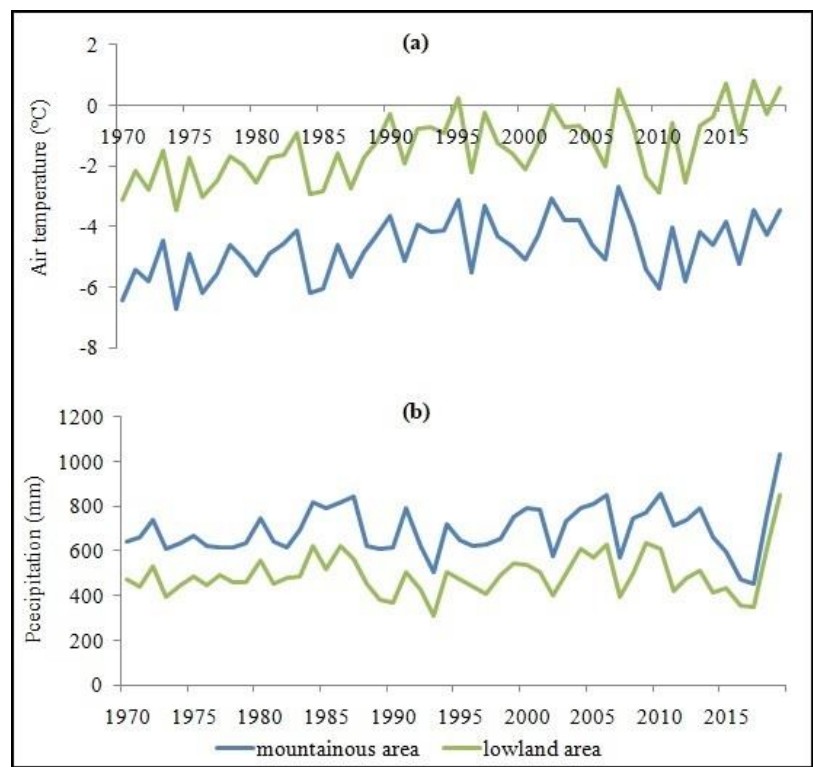

**Figure 3.** Interannual dynamics of air temperature (**a**) and precipitation (**b**) in the mountainous and lowland parts of the Iya River basin over the period 1970–2019.

The annual runoff from the mountainous part of the catchment is 59% of the runoff at the Tulun gauge, and 63% over the summer season. Thus, the annual runoff coefficient at the Arshan and Tulun gauges is 0.79 and 0.58, respectively.

The construction of a model river network and division of the catchment into 21 HRUs, with an average area of about 700 km², which are the calculation units of the model, were carried out at the stage of spatial schematization of the basin according to the HYDRO1k DEM data (Figure 1). The hydrological model of the Iya River basin was calibrated in a continuous mode for a much longer and more modern period of 1970–2019 than, for example, in [27].

Most of the parameters of the ECOMAG model for the Iya River basin were set a priori based on HWSD and GLCC. Soil parameters such as bulk density, porosity, field capacity, wilting point, and hydraulic conductivity were determined for each soil type using pedotransfer functions and soil grain-size distribution data. The model used the coefficients of snowmelt and infiltration, soil moisture evaporation, and freezing for various

LULC types. However, the range of variation of some parameters of the model was quite wide. The parameters of soil and LULC types for the entire river basin, for example, horizontal and vertical hydraulic conductivity of soil types, snowmelt, and evaporation coefficients of LULC types, etc. were calibrated, in contrast to the parameters of HRUs. Any HRU had its own set of soil and LULC types which defined the parameters of the model. It is important to emphasize some of the features of manual calibration of the ECOMAG model. First, the values of the key parameters of the land surface (elevation, soil and LULC) were the initial values for calibration, and the task was to find the optimal parameters near these initial values. Secondly, the calibration was organized in order to preserve the relationship between the values of a certain spatially distributed parameter of the soil or LULC type in the catchment. Thus, as a result of calibration, a set of spatially distributed parameters for the entire basin was determined using a combination of soil and LULC types.

The correspondence between the observed and simulated daily discharge was determined at the Arshan and Tulun gauges using the KGE, PBIAS, and RSR criteria. Since the model was used to assess the conditions for the formation of floods in the Iya River basin, its verification was carried out by calculating the error of the maximum daily discharge at the considered gauges.

In addition, taking into account the socioeconomic consequences of floods in the town of Tulun, the hydrological model was verified for this gauge according to the observed discharges exceeding the critical water level 455.91 m above the Baltic Height System (BHS) and 1 m below, at which residential buildings are flooded. The dependence of the discharge on the water level using daily data for the period 2008–2017 without ice phenomena, as well as the annual peak discharge for the entire observation period (since 1936), was obtained to determine the discharge corresponding to critical water levels, as well as to reconstruct the runoff in 2019 according to water levels. Thus, the discharges of 1100 m$^3$ s$^{-1}$ and 700 m$^3$ s$^{-1}$ were calculated as corresponding to the water levels of 455.91 m and 454.91 m BHS. Then, observed discharges exceeding these values for the period 1970–2019 were considered as two samples, for which statistical criteria were determined to match the simulated flow. The results of model testing both the daily discharge in a continuous mode and the annual peak discharge, as well as discharges exceeding critical water level threshold for a long-term period, were recognized as successful under the values of the statistical criteria of compliance of the observed and simulated runoff KGE $\geq$ 0.70, $|$PBIAS$| \leq$ 15% and RSR $\leq$ 0.60 [28]. Thus, a step-by-step procedure for testing the runoff generation model was applied to reproduce various indicators of river flow, similar to that described in [29,30]. A detailed analysis of the precipitation and streamflow regime during the highest floods in the Iya River basin during the observation period was carried out based on observed data and simulation results. The main factors of flood generation were identified according to the methodology in the Amur and Lena River basins [17,31].

### 2.2. Climate Change and River Runoff

Changes in the peak discharge, runoff during the period of exceeding the critical water level threshold, and its duration over the summer, as well as the maximum 5-day precipitation, were calculated in order to assess the impact of climate change on the Iya River runoff over the historical 50-year period using the hydrometeorological stations' data by dividing it into two equal periods: 1970–1994 and 1995–2019.

Estimation of future changes in the high flow of the Iya River was carried out based on the hydrological model and the AOGCM HadGEM2-ES of the Met Office Hadley Center, UK. The initial spatial resolution of the atmospheric block of the model is 1.875° by 1.25°, including 38 vertical levels, and the resolution of the oceanic block is 1° by 1° and 40 vertical levels [32]. Such a spatial resolution of AOGCMs makes it possible to use their output data for calculating the water regime of only large rivers. The bias-corrected data from the AOGCM HadGEM2-ES of the Inter-Sectoral Impact Model Intercomparison Project (ISIMIP https://www.isimip.org/ accessed on 12 March 2021), initiated by the Potsdam Institute

for Climate Research, were used to calculate future runoff changes in the Iya River basin. The AOGCM output data were downscaled to a grid of 0.5° by 0.5°, and the bias-correction procedure was carried out using the ERA reanalysis data (available online accessed on 12 March 2021 https://www.ecmwf.int/en/forecasts/datasets/reanalysis-datasets/era-interim), as well as the global databases of the Climate Research Unit (available online accessed on 12 March 2021 https://lr1.uea.ac.uk/cru/data) and the Global Precipitation Climatology Center (available online accessed on 12 March 2021 https://climatedataguide.ucar.edu/climate-data/gpcc-global-precipitation-climatology-centre) in order to improve the accuracy of reproduction of the intra-annual variation of meteorological values [33,34]. According to [35], the use of HadGEM2-ES output data for the river basins of Eastern Siberia and the Far East makes it possible to most effectively (in the framework of the ISIMIP) reproduce meteorological parameters in comparison with the meteorological station data. Studies of the impact of climate change on river runoff for catchments located in Asia, Europe, North and South America, Africa, and Australia were carried out based on the synthesis of regional hydrological models and AOGCMs in the framework of the ISIMIP (e.g., reviews in [36,37]).

To assess the change in hydrological characteristics for the future of the 21st century, it is necessary to identify the efficiency of the hydrological model, which was previously verified in the observational data, for reproducing the runoff using the AOGCMs output data over the historical period. Since the period under consideration in this article is from 1970 and historical data on the CMIP5 models are available until 2005, 1970–1999 was chosen as the reference period, in accordance with the recommendations of the World Meteorological Organization. The mean peak discharge over the summer, runoff during the period of exceeding the critical water level threshold, and its duration, as well as the maximum 5-day precipitation were calculated for the period 1970–1999 using the observational dataset and the HadGEM2-ES output data. After that, using the runoff generation model (without changing the calibration parameters), we calculated the changes in these hydrometeorological characteristics for the 30-year period of 2021–2050 and the end of the 21st century (2070–2099) relative to the period of 1990–2019.

## 3. Results and Discussion

### 3.1. Testing the Runoff Generation Model

The results of model calibration in continuous mode over a 50-year period since 1970 showed the following agreement between the observed and simulated daily discharges at the Arshan and Tulun gauges: KGE of 0.70 and 0.84, PBIAS of −9% and 8%, RSR of 0.59 and 0.51, respectively. Verification of the model for the accuracy of calculating the annual peak discharge at the gauges for the same period showed that PBIAS at the Arshan and Tulun gauges was −11% and −3%, and the correlation coefficient (R) was 0.76 and 0.97, respectively. Statistical criteria KGE and PBIAS for discharges exceeding 1100 m$^3$ s$^{-1}$ at the Tulun gauge were 0.91 and −1%, while the criteria for discharges exceeding 700 m$^3$ s$^{-1}$ were 0.88 and −2%, respectively (Table 3).

**Table 3.** Model performance of the Iya River at the Tulun gauge over the period 1970–2019.

| Calibration | | | Verification | | | | | |
|---|---|---|---|---|---|---|---|---|
| daily discharge | | | annual peak discharge | | | hazardous high flow | | |
| KGE | PBIAS, % | RSR | R | PBIAS, % | RSR | KGE | PBIAS, % | RSR |
| 0.84 | 8 | 0.51 | 0.97 | −3 | 0.30 | 0.91 | −1 | 0.41 |

Long-term daily flow duration curve (FDC) of the Iya River at Tulun gauge over the period 1970–2019 was plotted as an additional performance of the runoff generation model (Figure 4). The value of RSR for this FDC was 0.28.

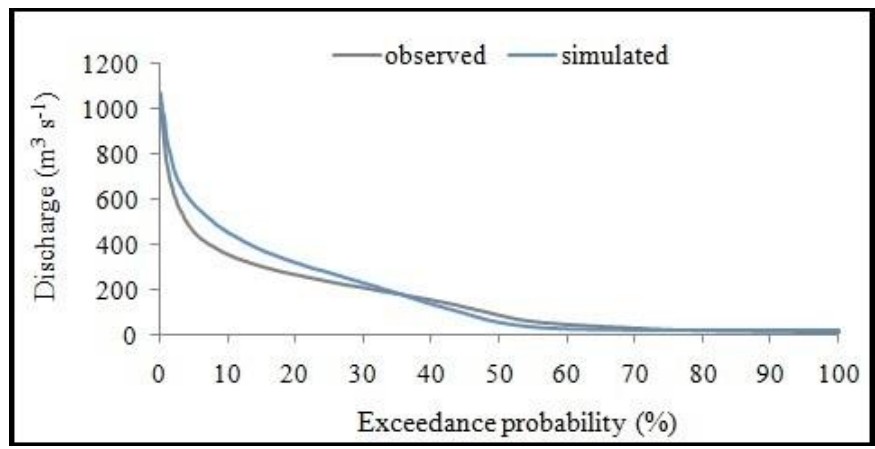

**Figure 4.** Long-term daily FDC of the Iya River at Tulun gauge over the period 1970–2019.

To represent the performance of the water balance in the hydrological model, a graph of the interannual dynamics of the observed and simulated runoff coefficients of the Iya River at Tulun gauge over the period 1970–2019 is plotted (Figure 5).

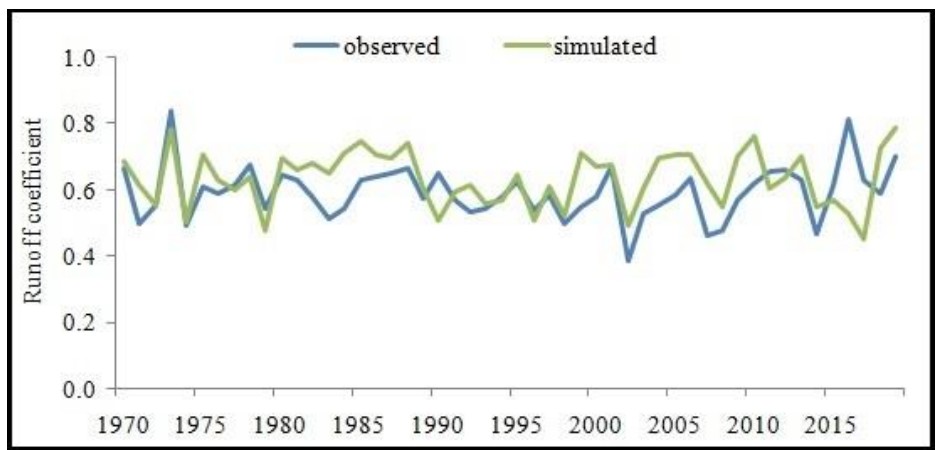

**Figure 5.** Interannual observed and simulated runoff coefficients of the Iya River at Tulun gauge over the period 1970–2019.

Given the study of the high runoff of the Iya River over the summer period, the hydrological model is most sensitive to the calibration parameters associated with horizontal and vertical hydraulic conductivity of soil types, as well as evaporation of LULC types (Table 4).

**Table 4.** Calibration parameters of the ECOMAG model for the Iya River basin.

| Parameter | Dimension | Value |
| --- | --- | --- |
| Coefficient for horizontal hydraulic conductivity of the topsoil layer | dimensionless | 10 |
| Coefficient for vertical hydraulic conductivity of soil type | dimensionless | 15 |
| Evaporation coefficient of LULC type | dimensionless | 0.35 |
| Baseflow of HRUs | mm day$^{-1}$ | 0.11 |
| Precipitation gradient | mm 100 m$^{-1}$ | 4 |
| Air temperature gradient | °C 100 m$^{-1}$ | −0.6 |
| Air temperature for transformation of precipitation phase | °C | 0.3 |
| Snowmelt air temperature | °C | 0.0 |
| Snowmelt intensity for LULC types | mm °C day$^{-1}$ | 0.28 |

Successful results of testing the model both for daily discharge in a continuous mode and for the annual peak discharge, and discharges exceeding the critical water level

threshold over a long-term period, allow it to be used to quantitatively determine the factors of flood formation in the Iya River basin and the impact of current and future climate changes on river runoff in 21st century.

*3.2. Flood Formation Factors on the Iya River*

During the observation period of 1936–2019, the highest annual peak discharge of the Iya River is characteristic over the summer period: 12 events in June, 49 events in July, 23 events in August. The average long-term date of annual peak discharge is 20 July. Since the air temperature in the catchment becomes positive in the third decade of April, annual peak discharge of the Iya River is associated with liquid precipitation, and not with snowmelt or possible ice-dams in spring.

During the observation period at the Tulun gauge, there were four catastrophic floods in 1980, 1984, 2006 and 2019 with daily peak discharges exceeding 2000 $m^3 s^{-1}$. The maximum recorded daily peak flood until 2019 was 4040 $m^3 s^{-1}$ on 23 July 1984 (measured 4100 $m^3 s^{-1}$). On 29 June 2019, the daily water level was 2.3 m higher than the previous maximum and equal to 462.22 m BHS (exceeding the critical level by 6.3 m and the protective dam of the town of Tulun by 3.3 m). Extrapolation of the dependence of the discharge on the water level exceeding those observed before 2019 at the Tulun gauge leads to an overestimation of the runoff by reason of destruction of the ratio between the mean velocity of streamflow and the channel cross section area at the gauge due to shallow areas near the river banks at a water level exceeding 459 m BHS, as well as additional resistance of floodplain vegetation, including shrubs and trees. In this regard, and due to the lack of runoff measurements, the peak discharge of 2019 was calculated based on the channel cross section area at the observed water level of 2019, and the mean velocity of streamflow measured for the peak discharge of 1984. The calculated daily peak discharge in 2019 was 5930 $m^3 s^{-1}$. However, it should be noted that, at the water level in 2019, the mean velocity of streamflow was higher than the value of 2.66 $m s^{-1}$ measured in 1984; but, taking into account the smaller increase in the velocity of streamflow at the water level exceeding 459 m BHS due to the gentle bank of the river with shrubby vegetation, these errors in determining the peak discharge can be taken as compensating each other. In addition, the federal highway R255–Siberia, which crosses the wide floodplain of the Iya River in the town of Tulun, and the Trans-Siberian Railway, located along the river downstream, contribute to an increase in the flooding of the urban area and a decrease in the longitudinal slope of the water surface and, thereby, the velocity of streamflow. Thus, the value of the daily peak discharge of 5930 $m^3 s^{-1}$ is approximate and is calculated under the conditions of the available data.

The results of runoff modeling at the Tulun gauge during the floods of 1980, 1984, 2006, and 2019 are presented in Table 5 and Figure 6. Considering the discharges during these floods as a single sample (about 4 months), the runoff generation model reproduces the daily flow at the Tulun gauge with the criteria KGE of 0.78, PBIAS of 11%, and RSR of 0.33, and at the Arshan gauge with KGE of 0.80, PBIAS of −1%, and RSR of 0.46. KGE was 0.82, PBIAS was 2%, and RSR was 0.38 for discharges exceeding the critical water level threshold at the Tulun gauge.

**Table 5.** Evaluation of the correspondence between observed and simulated discharges at Tulun gauge for the floods in 1980, 1984, 2006 and 2019 using the KGE, PBIAS and RSR criteria ($Q_1$–discharges for the entire period of flood, $Q_2$–discharges above the critical water level).

| Criteria | 1980 | | 1984 | | 2006 | | 2019 | |
|---|---|---|---|---|---|---|---|---|
| | $Q_1$ | $Q_2$ | $Q_1$ | $Q_2$ | $Q_1$ | $Q_2$ | $Q_1$ | $Q_2$ |
| KGE | 0.74 | 0.71 | 0.60 | 0.53 | 0.86 | 0.65 | 0.79 | 0.78 |
| PBIAS, % | 13 | 3 | 24 | 10 | 6 | −3 | 10 | 1 |
| RSR | 0.37 | 0.54 | 0.50 | 0.53 | 0.48 | 0.79 | 0.26 | 0.39 |

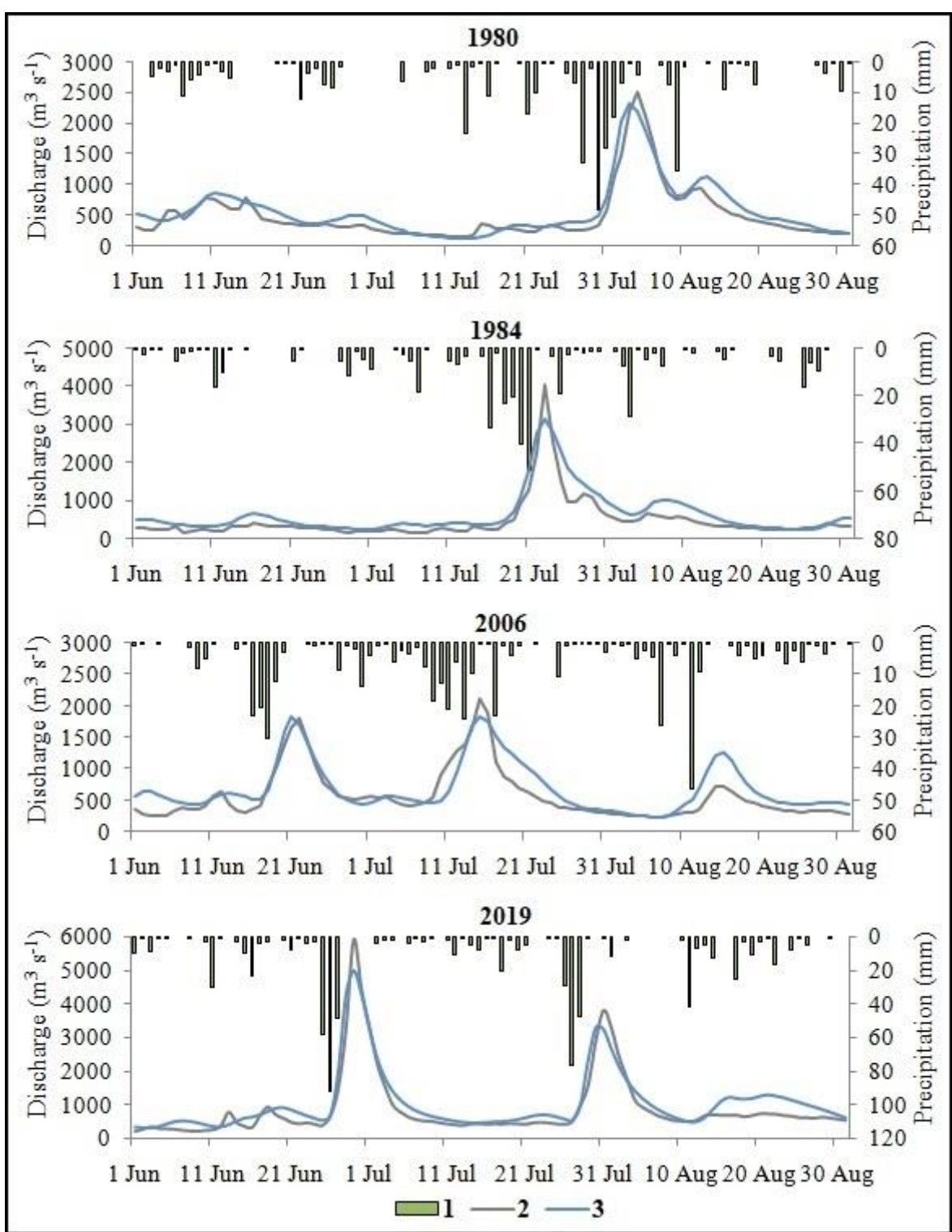

**Figure 6.** Averaged-basin precipitation (1), observed (2), and simulated (3) discharges of the Iya River at Tulun gauge over summers in 1980, 1984, 2006 and 2019.

Despite the satisfactory model results of runoff generation during the four considered catastrophic floods, it should be noted that the hydrological model underestimated the daily peak discharge at the Tulun gauge by an average of 15%, with a slight overestimation of the runoff per floods, which is generally typical for such models in mountainous areas using measured precipitation data. At the same time, the reproduction of the date of flood

peak was successful in three cases out of four (1984, 2006, 2019); in 1980 the simulated value was earlier than the observed one by one day.

The 1980 and 1984 floods had one wave in the first decade of August and the third decade of July, respectively. The number of days with discharges exceeding the critical water level threshold at the Tulun gauge was seven in 1980, and six in 1984. The floods of 2006 and 2019 had two waves, separated by two and three weeks, respectively. In 2006, the first wave was at the beginning of the third decade of June, and the second wave was in mid-July. In 2019, the first flood wave was in late June–early July, and the second wave was in late July–early August (Figure 7). Moreover, the peak discharge in 2006 was during the second wave of flooding, and the peak discharge in 2019 was during the first wave. The number of days with discharges exceeding the critical water level threshold at the Tulun gauge was four and seven days in 2006, and seven days for each wave in 2019.

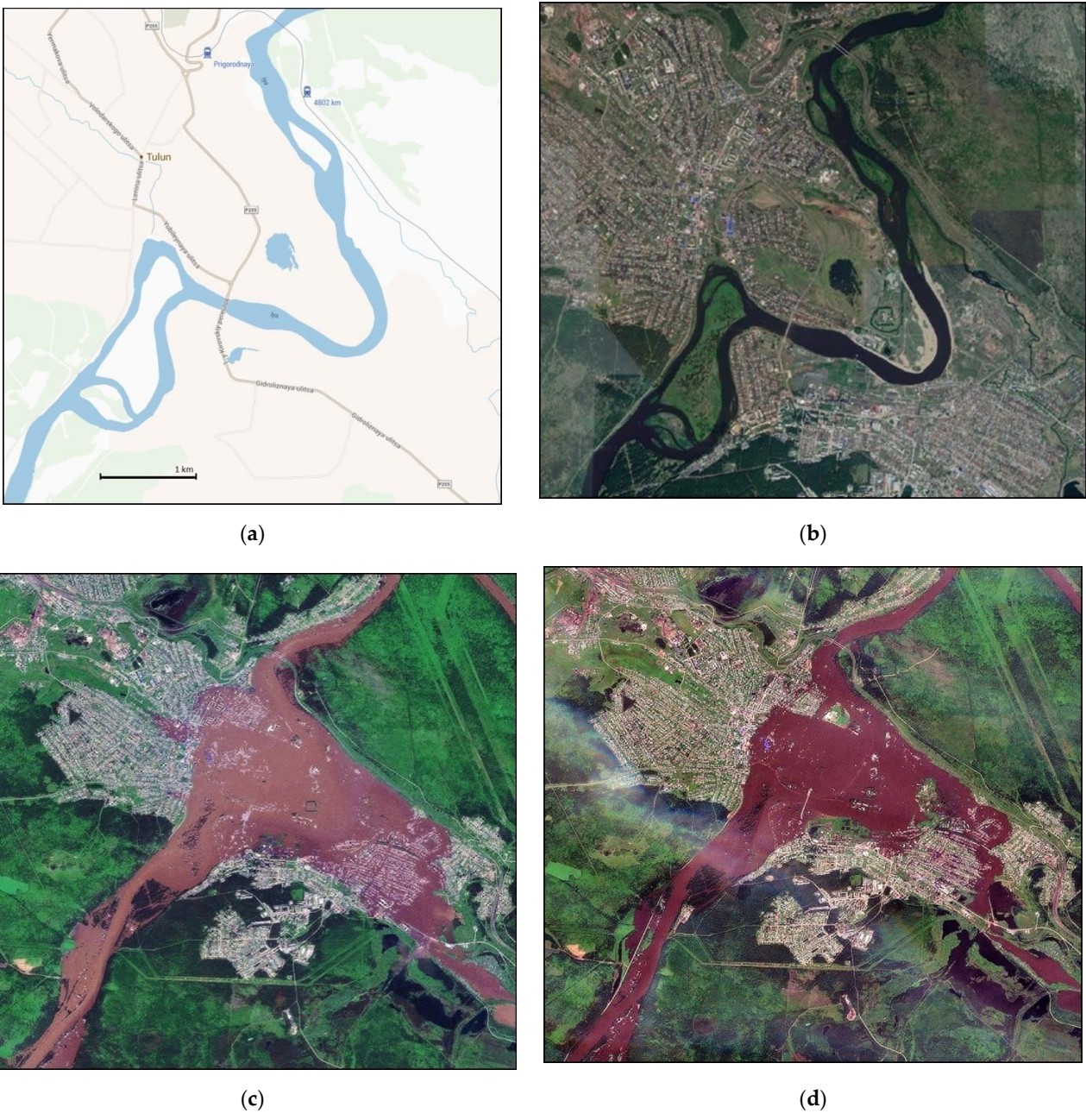

(a)

(b)

(c)

(d)

**Figure 7.** Schematic map of the town of Tulun (**a**), as well as satellite images of the territory during the low-water period (**b**), 29 June (**c**), and 31 July (**d**), 2019 according to the Sentinel-2A satellite.

Since the seasonal snow cover in the second half of June is not typical in the Iya River basin, snowmelt runoff cannot be the reason for the generation of these floods. The version with the spread of the backwater from the Bratsk reservoir is also false, because the altitude difference between the water level in the reservoir and at the Tulun gauge is about 50 m. The hypotheses about additional glacial runoff or the catastrophic impact of deforestation on river flow have not been confirmed either.

It should be noted that the relief conditions contribute to the regular flooding of the town of Tulun: the location of a significant part of the urban area in a wide-floodplain meander scroll and lowlands due to the confluence of the Azey and Tulunchik rivers into the Iya River. In addition, the anthropogenic impact, contributing to the increase in the flooded area, is manifested as a result of the construction of the federal highway R255– Siberia and the Trans-Siberian Railway. The analysis of natural factors of flood generation is presented below.

The main natural factors for the generation of floods on the Iya River were extreme liquid precipitation and previous soil waterlogging due to a number of heavy rains. The influence of these factors on the floods is considered below. At the same time, the river runoff was estimated during the period when the water levels at the Tulun gauge were above the critical value, and the mean monthly precipitation was calculated for the period 1970–1999. The runoff generation model was used to calculate the water saturation of the topsoil layer (0.5 m) for each of 21 HRUs of the Iya River basin on the date before extreme precipitation. Then, averaged-basin preflood coefficient of water-saturated topsoil was estimated in relation to the field capacity of soil types. The closer this value is to one, the higher the water saturation of the soil and the lower the possible losses for infiltration. To verify the model by soil moisture, detailed measurement data are required, since this is an extreme variable in the catchment area, and the calculation of the averaged-basin soil moisture based on the interpolation of measurement data at several points will lead to large error. Taking into account the use of a calibrated process-based model, which quite effectively reproduces the high flow of the Iya River, it was assumed that the simulated soil moisture values can be used to analyze the formation conditions of floods.

In the Iya River basin in July 1980, the mean monthly precipitation was exceeded by 20% for seven days, with maximum values on 28 and 30 July. Heavy rains on 13, 16, 21 and 22 July contributed to the increased preflood water saturation of the soil. The averaged-basin preflood coefficient of water-saturated soil before extreme precipitation was 0.87, and the runoff coefficient of flood-forming precipitation was 0.67 (Table 6).

**Table 6.** The calculated values of the averaged-basin preflood coefficient of water-saturated soil and the runoff coefficient of flood-forming precipitation for the period of exceeding the critical water level threshold at Tulun gauge in 1980, 1984, 2006, and 2019.

| Characteristics | 1980 | 1984 | I Wave 2006 | II Wave 2006 | I Wave 2019 | II Wave 2019 |
|---|---|---|---|---|---|---|
| Preflood coefficient of water-saturated soil | 0.87 | 0.93 | 0.94 | 0.91 | 1.00 | 0.95 |
| Runoff coefficient of flood-forming precipitation | 0.67 | 0.83 | 0.73 | 0.75 | 0.88 | 0.81 |

In July 1984, the mean monthly precipitation was exceeded by 40% for six days, with maximum values on 20 and 21 July; precipitation in the lowland part of the Iya River basin on these days exceeded precipitation in the mountainous part by 1.5 times. Heavy rains fell before the flood occurred on 28 June, and 1, 7 and 12 July. The averaged-basin preflood coefficient of water-saturated soil before extreme precipitation was 0.93, and the runoff coefficient of flood-forming precipitation was 0.83.

In June 2006, the mean monthly precipitation was exceeded by 15% for four days, with a maximum value on 18 June, and precipitation in the lowland part of the Iya River

basin exceeded precipitation in the mountainous part by 30%. Before that, heavy rains occurred on 29 May and 10 June. The averaged-basin preflood coefficient of water-saturated soil before extreme precipitation was 0.94, and the runoff coefficient of flood-forming precipitation was 0.73. Before the second main flood wave, there was 83% of the mean monthly precipitation for seven days in July, with maximum values on 11 July (2.5 times more in the mountainous part of the catchment) and 13 July (1.5 times more in the lowland part). The averaged-basin preflood coefficient of water-saturated soil before extreme precipitation was 0.91, and the runoff coefficient of flood-forming precipitation was 0.75.

Precipitation on 25–27 June 2019 in the Iya River Basin was more than 2.5 the monthly average, with a maximum value on 26 June (almost equal between the mountainous and lowland parts of the catchment). The averaged-basin preflood coefficient of water-saturated soil was equal to one, due to heavy rains on 3, 11, 15 and 16 June. Such significant precipitation over several days and being relatively evenly distributed over the catchment, taking into account the soil waterlogging, formed a catastrophic flood on 29 June, with the highest discharge over the observation period. The runoff coefficient of flood-forming precipitation was 0.88, i.e., the highest value among the considered floods. Three weeks later, at the end of July 2019, the mean monthly precipitation was exceeded by 40% for seven days, with maximum values on 27 and 28 July (almost equal between the mountainous and lowland parts of the catchment). The preflood soil moisture was high (the averaged-basin coefficient of water-saturated soil was 0.95), due to the previous wave of flooding and heavy rains on 12, 15, 18 and 20 July. The runoff coefficient of flood-forming precipitation was 0.81.

### 3.3. Impact of Climate Change on the High Flow of the Iya River

The occurrence of anomalous synoptic events is associated with current climate changes, causing an increase in the number and power of cyclones in the Northern Hemisphere, the frequency of periods with intense precipitation, and the scale of droughts [38]. The physical mechanisms of the formation of regional weather and climatic anomalies are extremely complex and depend on both the anthropogenic component of global warming and natural quasi-cyclic processes. In recent decades, extreme hydrological events in Eastern Siberia and the Far East are often associated with blocking anticyclones [39,40]. This effect was also observed during the formation of the 2019 flood on the Iya River. The meridional blocking process over Eastern Siberia from the Sayan Mountains to the Arctic Ocean was before the first flood wave at the end of June, and it was over the North of the Irkutsk Region and the Trans-Baikal Territory before the second flood wave at the end of July. The results of calculations of changes in the current and future high flow of the Iya River in the 21st century are presented below.

The main part of the the Iya River runoff is observed during the summer, from June to August. To assess the change in the high flow at the Tulun gauge over the 50-year period of 1970–2019, calculations were made for 1995–2019 compared to 1970–1994 according to the characteristics: the seasonal peak discharge and runoff during the period of exceeding the critical water level threshold and its duration, as well as the averaged-basin maximum 5-day precipitation as an indicator of the formation of high river flow. The ratio of the maximum 5-day precipitation and the peak discharge during the summer confirms the possibility of its use: the correlation coefficient between these characteristics for the period 1970–1994 was 0.88, and it was 0.92 for the period 1995–2019. The peak discharge and the maximum 5-day precipitation have increased by 2% and 1%, respectively, over the past 25 years, i.e., an increase in the frequency of extreme precipitation due to climatic changes has not been determined, despite the maximum daily discharge recorded in 2019. However, the runoff during the period of exceeding the critical water level threshold increased by 1.8 times, and its duration increased by 1.5 times. This is mainly due to the long-period floods in 2006 and 2019.

The increase in the monthly air temperature was 1.2 °C for each summer month for the period 1995–2019, relative to 1970–1994 (Table 7). Over the past 25 years, there has

been an increase in June precipitation by 26%, in contrast to July and August, for which the tendency to change in precipitation is not typical. In the absence of a significant change in the Iya River runoff in June by 3%, this fact indicates an increase in soil moisture and is one of the reasons for the formation of catastrophic floods in the event of extreme precipitation.

**Table 7.** Changes in hydrometeorological characteristics (T—air temperature, P—precipitation, Q—discharge) in the Iya River basin over the summer period 1995–2019 relative to 1970–1994.

| Period | 1970–1994 | | | 1995–2019 | | |
|---|---|---|---|---|---|---|
| **Characteristics** | **T, °C** | **P, mm** | **Q, m$^3$ s$^{-1}$** | **ΔT, °C** | **ΔP, %** | **ΔQ, %** |
| June | 12.0 | 75 | 345 | 1.2 | 26.2 | 3.2 |
| July | 14.2 | 125 | 346 | 1.2 | −2.2 | 0.3 |
| August | 11.6 | 114 | 329 | 1.2 | −1.3 | 8.3 |

Hazardous high flow and precipitation values in the Iya River basin, based on observational data and the AOGCM HadGEM2-ES output data, were calculated to compare accuracy over the 1970–1999 historical period. The mean seasonal peak discharge was 1000 m$^3$ s$^{-1}$ and 942 m$^3$ s$^{-1}$, the runoff during the period of exceeding the critical water level threshold was 3.3 and 2.7 km$^3$, its duration was 23 and 25 days, and the averaged-basin maximum 5-day precipitation was 58 and 53 mm, respectively. The correlation coefficient between the observed daily or annual peak discharges and those calculated from the AOGCMs data is weak, since the AOGCMs are not focused on reproducing these characteristics. The averaged-basin (up to the Tulun gauge) maximum daily precipitation, calculated from the meteorological station data and the AOGCM HadGEM2-ES output data, was 31 and 28 mm over the period 1970–1999, and the highest precipitation was 55 and 56 mm, respectively. If we evaluate the changes in the maximum daily and 5-day precipitation according to the RCP-scenarios, then their values were 70% and 77%, relative to the values according to the observations in 2019, i.e., the local maximum precipitation is less due to the computational grid of 0.5°. Therefore, the simulated discharges using the hydrological model are similar, i.e., AOGCMs can estimate future changes over long-term period.

Thus, a satisfactory result of the accuracy of calculating hydrometeorological characteristics was obtained for the historical period 1970–1999, and this approach for assessing the change in the high flow of the Iya River using the regional runoff generation model and the AOGCM output data was applied for calculations over the future period, until the end of 21st century. In the CMIP5 models, the calculations over the future period are performed according to the scenarios of socioeconomic development, which are expressed in the level of greenhouse gas emissions under the RCP-scenarios. An important stage of this study was to assess which of the four scenarios (RCP 2.6, RCP 4.5, RCP 6.0, RCP 8.5) the current change in hydrometeorological characteristics in the Iya River basin since 2006, is most consistent. As a result, the RCP 6.0 scenario is the most appropriate. A similar conclusion was obtained for a number of other large river basins in Eastern Siberia [35,41]. In this regard, calculations for the 21st century were carried out according to the RCP 6.0 scenario, as well as the RCP 2.6 scenario to estimate the high flow change of the Iya River, with the largest reduction in greenhouse gas emissions compared to the RCP 6.0 scenario.

According to the AOGCM output data, daily meteorological characteristics (air temperature and humidity, precipitation) were used as the boundary conditions in the runoff generation model of the Iya River. Thus, daily discharges were calculated until the end of the 21st century, according to the RCP 2.6 and RCP 6.0 scenarios. Then, the calculations of the seasonal peak discharge, runoff during the period of exceeding the critical water level threshold and its duration, as well as the averaged-basin maximum 5-day precipitation were carried out for the 30-year periods of 2021–2050 and the end of the 21st century (2070–2099). The period 1990–2019 was considered as a reference period for calculating the long-term average values of these characteristics based on observational data. As a result,

the anomalies (changes) of the high flow of the Iya River at the Tulun gauge and averaged-basin precipitation were estimated according to the RCP 2.6 and RCP 6.0 scenarios, relative to the 1990–2019 values (Table 8).

**Table 8.** Anomalies of the seasonal peak discharge, hazardous high flow and its duration, as well as the maximum 5-day precipitation over the summer in the Iya River basin for the period 2021–2050 and the end of the 21st century (2070–2099), relative to 1990–2019.

| Period | 2021–2050 | | 2070–2099 | |
|---|---|---|---|---|
| Characteristics | RCP 2.6 | RCP 6.0 | RCP 2.6 | RCP 6.0 |
| Peak discharge, % | 3 | 3 | −16 | −23 |
| Hazardous high flow, % | −8 | 15 | −41 | −20 |
| Duration of hazardous high flow, day | 12 | 18 | −3 | 4 |
| Maximum 5-day precipitation, % | 13 | 11 | −5 | 4 |

As a result, the seasonal peak discharge can be expected to increase by 3% (relative to 980 m$^3$ s$^{-1}$ for the period 1990–2019), and the maximum 5-day precipitation by 11–13% (relative to 58 mm) during the period 2021–2050, according to the RCP 2.6 and RCP 6.0 scenarios. The runoff during the period of exceeding the critical water level threshold will increase by 15% under the RCP 6.0 scenario and decrease by 8% under the RCP 2.6 scenario (relative to 5.4 km$^3$). The duration of hazardous high flow will increase by 12 and 18 days (relative to 31 days) according to the RCP 2.6 and RCP 6.0 scenarios, respectively.

During the period 2070–2099, it is possible to expect a decrease in the peak discharge by 16–23% according to the RCP 2.6 and RCP 6.0 scenarios, the maximum 5-day precipitation by 5% under the RCP 2.6 scenario, and its increase by 4% under the RCP 6.0 scenario. Reduction of future hazardous high flow of the Iya River by 20–41% is determined according to the RCP 2.6 and RCP 6.0 scenarios. At the same time, the duration of hazardous high flow will decrease by three days according to the RCP 2.6 scenario, and increase by four days according to the RCP 6.0 scenario.

## 4. Conclusions

The semi-distributed process-based ECOMAG model was applied to the Iya River basin using global databases of land surface parameters and hydrometeorological observations. Successful model testing results were obtained for the daily discharge in continuous mode, the annual peak discharge, and discharges exceeding the critical water level threshold over the multiyear period of 1970–2019 using the KGE, PBIAS and RSR statistical criteria. Considering the discharges during four of the highest floods as a single sample (about four months), the runoff generation model reproduces the daily flow at the Tulun gauge with the criteria KGE of 0.78, PBIAS of 11%, and RSR of 0.33. KGE was 0.82, PBIAS was 2%, and RSR was 0.38 for discharges exceeding the critical water level threshold. The use of a process-based model for the Iya River in 2019, similar to that presented in this article, would allow us to calculate the flood with a greater forecast lead time and use this time to evacuate the population and minimize socioeconomic damage.

The analysis of natural factors of flood formation in the Iya River basin showed that, first of all, floods were caused by a series of extreme precipitation over several days exceeding the mean monthly precipitation on waterlogged soil (preflood coefficient of water-saturated soil is more than 0.9). The maximum recorded daily peak discharge was formed at the end of June 2019 under the conditions of the highest averaged-basin preflood coefficient of water-saturated soil, and 2.5 mean monthly precipitation for three days. In general, the runoff coefficient of an extreme hydrological event is determined not only by the previous soil moisture, but, of course, by the intensity of precipitation. Thus, the three maximum recorded daily peak discharge on 23 July 1984, 29 June, and 31 July 2019 were formed as a result of extreme precipitation over two or three days, either relatively evenly in the Iya River basin, or in the case of more precipitation in the lowland part of the catchment compared to the mountainous area.

The annual peak discharge of the Iya River at the Tulun gauge increased by 2% during the period 1995–2019 relative to 1970–1994, despite the maximum recorded daily peak discharge in 2019. According to [42], there were no statistically significant changes in the frequency of floods in the Lena and Yenisei River basins during the period 1960–2005. At the same time, the Iya River runoff at the Tulun gauge during the period of exceeding the critical water level threshold, when flooding of residential buildings occurs, increased by 1.8 times, and its duration increased by 1.5 times.

According to various RCP-scenarios of the future climate in the Iya River basin, there will be less change in the annual peak discharge or precipitation and more change in the hazardous flow and its duration, exceeding the critical water level threshold, at which residential buildings are flooded. Estimating the difference between the RCP-scenarios, in general, RCP 6.0 scenario is characterized by more extreme values of hazardous high flow and its duration, by about 20% more runoff, and a week longer. Despite a significant increase in the duration of hazardous high flow during the period 2021–2050, and almost unchanged by the period 2070–2099, the runoff decreases during this period for both the RCP-scenarios by the end of the 21st century and during the period 2021–2050 by the RCP 2.6 scenario, or increases by 15% according to the RCP 6.0 scenario. This effect is explained by the lower values of discharges exceeding 1100 $m^3\ s^{-1}$ in the future compared to the reference period of 1990–2019. This is mainly due to the extreme discharges during the two waves of the 2019 flood, which significantly increased the runoff of the historical period. Similar daily peak discharge has not been identified in the future; however, taking into account the increase in the duration of hazardous high flow in the Iya River basin, especially in the next thirty years, the author recommends organizing the optimization of flood protection structures near the town of Tulun.

The hydrological model presented in this article can be used both as a method for studying the floods that have occurred and for short-term forecasting of future floods based on forecast data from mesoscale weather models, as well as for calculations of possible long-term changes in both the climate and in the land surface parameters (for example, deforestation due to felling or fires), given the extreme importance of research in recurring floods for this region and the interest at the state level.

**Funding:** This research was funded by the Russian Science Foundation (grant 20-77-00077—part of the methodology for calculating changes in high flow and precipitation for a mountain catchment) and the Ministry of Science and Higher Education of the Russian Federation (grant MK-1753.2020.5— part of developing the runoff generation model, analysis of flood factors and climate change impact).

**Institutional Review Board Statement:** Not applicable.

**Informed Consent Statement:** Not applicable.

**Data Availability Statement:** The AOGCM output data are available on the ISIMIP ESGF server (data link https://esg.pik-potsdam.de/projects/isimip/ accessed on 12 March 2021). The data of hydrological modeling presented in this study are available on request from the author.

**Acknowledgments:** The author is grateful to the USGS and FAO for the DEM, soil and landuse data, as well as to the ISIMIP for the AOGCM output data and RIHMI-WDC for the meteorological station data used for hydrological modeling.

**Conflicts of Interest:** The author declares no conflict of interest.

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
