# Peer review of "Process-Based Modeling of the High Flow of a Semi-Mountain River under Current and Future Climatic Conditions: A Case Study of the Iya River (Eastern Siberia)"

_water, doi:10.3390/w13081042_

Round 1

Reviewer 1 Report

Ms. Ref. No.: water-1162041

Title: Process-based Modeling of the High Flow of a Semi-Mountain River under Current and Future Climatic Conditions: a Case Study of the Iya River (Eastern Siberia)

Journal: Water

General Comments

The authors have done a good amount of work to justify the implementation of semi-distributed process-based ECOMAG model to the Iya River basin by using several global datasets and future RCP scenarios to estimate the streamflow discharge. However, there are some points described below that have to be considered before publication. I believe that after duly addressing the comments authors can improve the quality of the manuscript substantially to make it more insightful.

-There is a need to clearly state the objective(s) of the study towards the end of the introduction.

- Discussion and Conclusion sections can be more rigorous with an objective base.

**Abstract**

 Line 10 make st in the superscript across all the places in the manuscript

Line 12 and 13 use the runoff and discharge term cautiously as they are two different processes over the catchment

Line 13-14 Rephrase the sentence

Line 16 superscript

Line 17 expand HadGEM2-ES

Line 19 RCP

Line 20 “one can expect not so much a change” the sentence is not properly formed please rephrase it

In the results part of the abstract there are no values so it would be great if author an add the the R2 and NSE values in this section and one novel point of the results.

**Introduction**

English editing is required in the entire manuscript however mostly in the introduction portion where I guess most of the sentences are not correctly formed. Also the literature review covered by the author in this section is far to narrow so here I would recommend using some of them mention below.

For example, authors have stated that the range of population growth and agricultural land area affected by a two fold increase in the frequency floods, respectively, varies by 15 and 7 times so this sentence can be supported by adding a recent study by Srivastava et al., 2020, where they have shown that agricultural land use heterogeneity tends to alters the flow regime and can cause the floods therefore I would recommend the authors to incorporate this recent study to build up and strengthen their introduction. Additionally, there are some more literature not limited to this which authors can benefit in supporting their paragraphs:

Srivastava, A., Kumari, N. & Maza, M. (2020). Hydrological Response to Agricultural Land Use Heterogeneity Using Variable Infiltration Capacity Model. Water Resour Manage 34, 3779–3794. https://doi.org/10.1007/s11269-020-02630-4

Kumari, N., Acharya, S.C., Renzullo, L.J., and Yetemen, O. (2019). Applying rainfall ensembles to explore hydrological uncertainty. 23rd International Congress on Modelling and Simulation (MODSIM2019), Canberra, Australia, pp. 1070-1076. https://doi.org/10.36334/modsim.2019.K14.kumari2

Line 38 replace the 15 and 7 times with 7 and 15 times in the order

Line 32 th should be in superscript

 Line 68-72 this entire paragraph should be shifted to the description of the study area as it more suits at that place.

 Line 78-81 Contradictory statement as many physically distributed models have shown good results. Need to rephrase it. I have big concern about the statement mention in these lines. Author have stated to use the physical based modelling in the data scarce regions, having limited hydrometerological observation. However I totally disagree with this statement for many reasons. Physically based modelling are data intensive, complex and sometimes does not produce better results. Conceptual models have been able to produce best results despite using less data and less data intensive, without their complexity it has been applied in many parts of the world such as Australia, Canada, and many part of the Asia. The esource platform in the Australia provides the detailed overview on the use of conceptual models vs physically based models. There is a lot of research out there on these topics. The author must review the recent literature to provide a clearer context for this work.  For this I would like to suggest authors to consider adding some of these recent literature that will be useful for the following manuscript in attributing/interpreting some of your statements in the introduction section.  Author can add a sentence “Although it can be argued that physically based hydrologic models could be the best choice for modeling rainfall-runoff process in data-scarce watersheds, the need of comprehensive comparison of the performance of various lumped conceptual model structures is strongly felt on regions with limited data availability (Paul et al., 2018; Srivastava et al., 2020; Darbandsari and Coulibaly, 2020).”

Srivastava, A., Deb, P., & Kumari, N. (2020). Multi-Model Approach to Assess the Dynamics of Hydrologic Components in a Tropical Ecosystem. Water Resources Management, 34(1), 327-341.

Paul, P. K., Kumari, N., Panigrahi, N., Mishra, A., & Singh, R. (2018). Implementation of cell-to-cell routing scheme in a large scale conceptual hydrological model. Environmental modelling & software, 101, 23-33.

Darbandsari, P., & Coulibaly, P. (2020). Inter-comparison of lumped hydrological models in data-scarce watersheds using different precipitation forcing data sets: Case study of Northern Ontario, Canada. Journal of Hydrology: Regional Studies, 31, 100730.

Line 86 what do you mean by semi-mountain rivers please elaborate. Also not mentioned in the entire study how orographic precipitation is used in the ECOMAG model as by looking at the index map it seems that there exists the elevation control.

**Materials and method**

Line 89 “Analysis of publications” where have the authors have shown this it is not clearly stated in the introduction

Line 91 mention source of the ECOMAG software, is it publicly accessible

Line 99 how do you define the criteria for the classification of the rivers please provide the details/

 Line 102 please be careful in using altitude and elevation

Mention the source of using HYDRO1k DEM with its spatial resolution

Line 110-112 provide a table for the sources of all the data obtained with it duration.

In the caption of the Figure 1 what do the “calculation units” mean I would suggest using index map. Caption is not well elaborated include the details of the map adequately.

There is very short description of the model provided in the manuscript which does not give clear idea of the model processes such as ET, runoff, infiltration. Therefore I would suggest authors to add a detailed description of all these processes with their governing equation used in the model for better understanding. Authors can take reference from the several studies also they can get an overall idea from the above studies.

Line 118 provide the source links

Line 119 please provide the more details such as resolution and date of collection of the image.

Figure 2 – it should be soil map , b) landuse/landcover map of

Line 129-130 It is recommended to show the timeseries plot of temperature and precipitation

Table 1: It’s a good way to represent hydrometerological datasets however I suggest to author to give convert Table 1 into graph.

Line 141-142 Two indicators are not sufficient to justify the results therefore I would suggest authors to add some more particular volume error, rmse etc.

Line 160-161 What is the basis of this range? Here I would suggest to go through the Moriasi et al., 2007 for model evaluation guidelines.

Moriasi, D. N., Arnold, J. G., Van Liew, M. W., Bingner, R. L., Harmel, R. D., & Veith, T. L. (2007). Model evaluation guidelines for systematic quantification of accuracy in watershed simulations. Transactions of the ASABE, 50(3), 885-900.

It is not clear from this paragraph which type of calibration have author used in their study for the simulation of model discharge.

Line 165-166 sentence need restructuring

Line 168-187 Enlist all the data sources and the citations.

Line 173 is there any specific reason to use this circulation model

**Results and Discussion**

Provide a table showing efficiencies for calibration and validation and with the calibrated model parameters.

From the results section I feel author have provided the hydrograph however I would recommend few things to make the results section more comprehensible.

Include flow duration curves for discharge for calibration and validation period similar to the findings of the obtained in Srivastava et al., 2018. They can add the description from this study. Also author can use the exceedance probability distribution to showcase the frequency distribution.

Srivastava, A., Sahoo, B., Raghuwanshi, N. S., & Chatterjee, C. (2018). Modelling the dynamics of evapotranspiration using Variable Infiltration Capacity model and regionally calibrated Hargreaves approach. Irrig Sci 36, 289–300 (2018). https://doi.org/10.1007/s00271-018-0583-y

Also it would be great if authors can show the water balance components obtained from the model for the basin to insights how the model works for each processes.

In the hydrographs please add the precipitation on the secondary axis in Figure 3.

**Conclusions**

Authors should mention in the conclusion where the models applied need to be improved or any suggestions for future research in terms of models' performance improvement and application

Author Response

Response to Reviewer 1 Comments

The author is very grateful to the reviewer for effective comments and suggestions.

General Comments

The authors have done a good amount of work to justify the implementation of semi-distributed process-based ECOMAG model to the Iya River basin by using several global datasets and future RCP scenarios to estimate the streamflow discharge. However, there are some points described below that have to be considered before publication. I believe that after duly addressing the comments authors can improve the quality of the manuscript substantially to make it more insightful.

-There is a need to clearly state the objective(s) of the study towards the end of the introduction.

- Discussion and Conclusion sections can be more rigorous with an objective base.

Response: Сomment on the purpose of the study was added to the introduction (Line 99-108 in the attached file water-1162041_editing.pdf).

Accordingly, the development of a process-based hydrological model with distributed parameters (the main method of this study) using a continuous long-term period of hydrometeorological observations from 1970 to 2019, including recent years, and its verification by the accuracy of calculating high runoff, will make it possible to carried out a comprehensive analysis of the formation conditions of the highest floods in within a unified methodological approach. This is one aspect of the purpose of this study. Another aspect is that the applied method of process-based hydrological modeling makes it possible to obtain the results of the influence of future climate changes on the physical transformation of the high flow of the Iya River, which may differ significantly from the historical period.

**Abstract**

Line 10 and Line 16 make st in the superscript across all the places in the manuscript

Corrected everywhere

Line 12 and 13 use the runoff and discharge term cautiously as they are two different processes over the catchment

Daily runoff replaced with daily discharge (Line 13)

Line 13-14 Rephrase the sentence «Quantitative estimates of the preflood coefficient of water-saturated soil and the runoff coefficient of flood-forming precipitation for 1980, 1984, 2006, and 2019 were obtained.»

The preflood coefficient of water-saturated soil and the runoff coefficient of flood-forming precipitation in the Iya River basin were calculated in 1980, 1984, 2006, and 2019. (Line 18-20)

Line 17 expand HadGEM2-ES

Added to abstract: Hadley Centre Global Environment Model version 2 – Earth System (Line 22-23)

Line 19 RCP

The acronym RCP moved from section 3.3 to abstract. (Line 26)

Line 20 “one can expect not so much a change” the sentence is not properly formed please rephrase it

The sentence “According to various Representative Concentration Pathways (RCP-scenarios) of the future climate in the Iya River basin, one can expect not so much a change in the maximum flow or precipitation, but rather a change in the runoff above the critical water level at which residential buildings are flooded, and its duration” was rephrased «According to various Representative Concentration Pathways (RCP-scenarios) of the future climate in the Iya River basin, there will be less change in the annual peak discharge or precipitation, and more change in the flow exceeding the critical water level threshold at which residential buildings are flooded, and its duration». (Line 25-29)

In the results part of the abstract there are no values so it would be great if author an add the the R2 and NSE values in this section and one novel point of the results.

Modeling of the high flow of the Iya River was carried out according to the Kling-Gupta efficiency (KGE) of 0.91, percent bias (PBIAS) of –1% and ratio of the root mean square error to the standard deviation of measured data (RSR) of 0.41. (Line 14-17)

**Introduction**

English editing is required in the entire manuscript however mostly in the introduction portion where I guess most of the sentences are not correctly formed. Also the literature review covered by the author in this section is far to narrow so here I would recommend using some of them mention below.

For example, authors have stated that the range of population growth and agricultural land area affected by a two fold increase in the frequency floods, respectively, varies by 15 and 7 times so this sentence can be supported by adding a recent study by Srivastava et al., 2020, where they have shown that agricultural land use heterogeneity tends to alters the flow regime and can cause the floods therefore I would recommend the authors to incorporate this recent study to build up and strengthen their introduction. Additionally, there are some more literature not limited to this which authors can benefit in supporting their paragraphs:

Srivastava, A., Kumari, N. & Maza, M. (2020). Hydrological Response to Agricultural Land Use Heterogeneity Using Variable Infiltration Capacity Model. Water Resour Manage 34, 3779–3794. https://doi.org/10.1007/s11269-020-02630-4

Kumari, N., Acharya, S.C., Renzullo, L.J., and Yetemen, O. (2019). Applying rainfall ensembles to explore hydrological uncertainty. 23rd International Congress on Modelling and Simulation (MODSIM2019), Canberra, Australia, pp. 1070-1076. https://doi.org/10.36334/modsim.2019.K14.kumari2

English edited throughout the manuscript.

The source of suggested research has been added. (Line 49)

Line 38 replace the 15 and 7 times with 7 and 15 times in the order

Corrected (Line 47-48)

Line 32 th should be in superscript

Corrected (Line 41)

Line 68-72 this entire paragraph should be shifted to the description of the study area as it more suits at that place.

The author agrees, the paragraph has been moved to the beginning of the Materials and methods section (Line 155-159)

Line 78-81 Contradictory statement as many physically distributed models have shown good results. Need to rephrase it. I have big concern about the statement mention in these lines. Author have stated to use the physical based modelling in the data scarce regions, having limited hydrometerological observation. However I totally disagree with this statement for many reasons. Physically based modelling are data intensive, complex and sometimes does not produce better results. Conceptual models have been able to produce best results despite using less data and less data intensive, without their complexity it has been applied in many parts of the world such as Australia, Canada, and many part of the Asia. The esource platform in the Australia provides the detailed overview on the use of conceptual models vs physically based models. There is a lot of research out there on these topics. The author must review the recent literature to provide a clearer context for this work.  For this I would like to suggest authors to consider adding some of these recent literature that will be useful for the following manuscript in attributing/interpreting some of your statements in the introduction section.  Author can add a sentence “Although it can be argued that physically based hydrologic models could be the best choice for modeling rainfall-runoff process in data-scarce watersheds, the need of comprehensive comparison of the performance of various lumped conceptual model structures is strongly felt on regions with limited data availability (Paul et al., 2018; Srivastava et al., 2020; Darbandsari and Coulibaly, 2020).”

Srivastava, A., Deb, P., & Kumari, N. (2020). Multi-Model Approach to Assess the Dynamics of Hydrologic Components in a Tropical Ecosystem. Water Resources Management, 34(1), 327-341.

Paul, P. K., Kumari, N., Panigrahi, N., Mishra, A., & Singh, R. (2018). Implementation of cell-to-cell routing scheme in a large scale conceptual hydrological model. Environmental modelling & software, 101, 23-33.

Darbandsari, P., & Coulibaly, P. (2020). Inter-comparison of lumped hydrological models in data-scarce watersheds using different precipitation forcing data sets: Case study of Northern Ontario, Canada. Journal of Hydrology: Regional Studies, 31, 100730.

In lines 78-81, the standard flow forecasting methods mean the use of such simple methods as the flow transformation method along the channel or various statistically dependencies, which are widely used in Roshydromet organizations in Russia. Of course, these methods lead to large forecast errors under climate change and, in particular, what happened when forecasting the 2019 flood on the Iya River, including taking into account the rare hydrometeorological station network in the catchment. Unfortunately, the methods of physically-based modeling using both lumped and spatially distributed models for forecasting river flow in Russia have been introduced very locally, for example, in the Amur and Volga River basins. The use of a process-based model for the Iya River in 2019, similar to that presented in this article, would allow to calculate the flood with a greater forecast lead time and use this time to evacuate the population and minimize socio-economic damage.

Clarification of the sentence on lines 78-81 has been added in the introduction “using such simple flow forecasting methods as the flow transformation along the channel or various statistically dependencies” (Line 95-96) and conclusions “The use of a process-based model for the Iya River in 2019, similar to that presented in this article, would allow to calculate the flood with a greater forecast lead time and use this time to evacuate the population and minimize socio-economic damage” (Line 622-625).

In addition, information has been added at the end of the introduction to clarify the general purpose of this study (Line 99-108).

Regarding the choice of a method for research the influence of possible climate changes on river runoff, physically-based models with distributed parameters still have an advantage over models with lumped parameters, since the former are characterized by a more complex and deep understanding of the parameters of the hydrological cycle in the catchment area due to the input data used, and they are more adapted to the calculations of river runoff, especially high runoff, in significantly different climatic conditions from the observation period. This has been confirmed by a number of highly cited papers, for example, in the ISIMIP project using various rainfall-runoff models (Hattermann et al, 2017; Huanf et al, 2017; Krysanova et al, 2018). However, the author added the sentence suggested by the reviewer (Line 113-117).

Hattermann, F.F., Krysanova, V., Gosling, S.N. et al. Cross‐scale intercomparison of climate change impacts simulated by regional and global hydrological models in eleven large river basins. Climatic Change 141, 561–576 (2017). https://doi.org/10.1007/s10584-016-1829-4

Huang, S., Kumar, R., Flörke, M. et al. Evaluation of an ensemble of regional hydrological models in 12 large-scale river basins worldwide. Climatic Change 141, 381–397 (2017). https://doi.org/10.1007/s10584-016-1841-8

Valentina Krysanova, Chantal Donnelly, Alexander Gelfan, Dieter Gerten, Berit Arheimer, Fred Hattermann & Zbigniew W. Kundzewicz (2018) How the performance of hydrological models relates to credibility of projections under climate change, Hydrological Sciences Journal, 63:5, 696-720, DOI: 10.1080/02626667.2018.1446214

Line 86 what do you mean by semi-mountain rivers please elaborate. Also not mentioned in the entire study how orographic precipitation is used in the ECOMAG model as by looking at the index map it seems that there exists the elevation control.

In terms of the catchment area, the Iya River belongs to the category of medium-sized rivers (with an area of ​​2,000 to 50,000 km2). The upper part of the Iya River basin up to the Arshan gauge belongs to the mountain type of rivers, taking into account the large values ​​of the longitudinal slope (about 0.38%) and, accordingly, the streamflow velocity, as well as the rocky bottom. In the section of the Iya River between the Arshan and Tulun gauges, the longitudinal slope is 0.09%, i.e. the value is similar for lowland rivers. However, if considering the catchment area to the Tulun gauge, then the Iya River can be classified as a semi-mountain river. Taking into account the large height difference between the source and the mouth of the river, as well as the location of meteorological stations at an altitude of up to 1000 m, it became necessary to determine the precipitation gradient of 40 mm per 100 m of elevation for the mountain part of the catchment under calibration of the ECOMAG model.

This information has been added to the Materials and Methods section (Line 164, 166-171 and 190-194).

**Materials and method**

Line 89 “Analysis of publications” where have the authors have shown this it is not clearly stated in the introduction

Corrected. Analysis of publications in the abstract and citation databases Web of Science and Scopus (Line 120-121)

Line 91 mention source of the ECOMAG software, is it publicly accessible

The ECOMAG model is not yet open-source, line 123 indicates the source for the patent of its author [24] Motovilov, Y. ECOMAG, Rospatent: Water Problems Institute, Russian Academy of Sciences, 2013.

Line 99 how do you define the criteria for the classification of the rivers please provide the details

Above, the author explained in detail the principle of classification of the Iya River. (Line 164, 166-171)

Line 102 please be careful in using altitude and elevation

Replaced by average elevation (Line 173, 178)

Mention the source of using HYDRO1k DEM with its spatial resolution

HYDRO1k DEM resolution is 1 km, the source has been added (Line 175-176)

Line 110-112 provide a table for the sources of all the data obtained with it duration.

Table 1 has been added. (Line 195)

Table 1. Information about meteorological stations, daily data of which are used as boundary conditions in the runoff generation model over the period 1970-2019.

Index of the World Meteorological Organization

Name

Latitude, ° N

Longitude, ° E

Altitude,

m a.s.l.

29894

Alygdzher

53.63

98.22

1031

30504

Tulun

54.6

100.6

487

30505

Kuitun

54.3

101.5

520

30507

Ikey

54.18

100.08

510

30605

Saram

53.3

101.2

622

In the caption of the Figure 1 what do the “calculation units” mean I would suggest using index map. Caption is not well elaborated include the details of the map adequately.

Figure 1 updated: Location of hydrometeorological stations and hydrologic response units of the runoff generation model in the Iya River basin (8230 – Arshan gauge, 8233 – Tulun gauge) (Line 180)

There is very short description of the model provided in the manuscript which does not give clear idea of the model processes such as ET, runoff, infiltration. Therefore I would suggest authors to add a detailed description of all these processes with their governing equation used in the model for better understanding. Authors can take reference from the several studies also they can get an overall idea from the above studies.

The author fully agrees that a detailed description of the processes of the applied hydrological model and its calibration should be added.

In a model schematization of a river basin, its surface is divided into subbasins (hydrological response units – HRUs) based on a digital elevation model and the structure of the river network. Modeling of hydrological processes at each HRU is performed for four levels: the topsoil layer, horizon of caliche, groundwater and prechannel flow. During the cold season, snow cover is added. Prechannel flow is formed after filling the depressions of the land surface due to the formation of excess water, which does not infiltrate and flows down the slopes of the catchment to the river network. As a result of infiltration in the soil, water moves in the aeration zone along a slope to the channel network or transforms into a groundwater zone. In the model, the subsurface and groundwater flow is described according to the Darcy equation, and the prechannel and stream flow is described by the kinematic wave equation. The total porosity in the soil aeration zone is divided into capillary and non-capillary zones. It is assumed that when soil moisture is less than the field capacity, all soil water is in the capillary zone, and when soil moisture is greater than the field moisture capacity, it is in the non-capillary zone. In conditions of high soil moisture, the actual evaporation is equal to the potential, and then it decreases linearly to zero as the soil moisture decreases to wilting point. Potential evaporation is estimated according to the Dalton method.

The snowmelt rate is calculated using the degree-day method. The phase transformation of precipitation depends on the air temperature. The evaporation of solid and liquid phases of snow is estimated using data on the air humidity deficit. It is assumed that vertical temperature profiles in snow, frozen and thawed soil differ insignificantly from linear ones, and moisture migration to the soil freezing front is insignificant. Under these conditions, the dynamics of the depth of freezing and thawing of the soil can be described by a system of differential equations. Infiltration of rain and melt water into frozen soil is calculated taking into account the effect of ice content in frozen soil on the hydraulic conductivity of the soil. A more detailed mathematical description of the flow generation processes in the ECOMAG model is presented by its author in (Motovilov et al, 1999; Motovilov, 2016).

Most of the parameters of the ECOMAG model for the Iya River basin are set a priori based on Harmonized World Soil Database (HWSD) and Global Land Cover Characterization (GLCC). Such soil parameters as bulk density, porosity, field capacity, wilting point, hydraulic conductivity are determined for each soil type using pedotransfer functions and soil grain-size distribution data. The model uses the coefficients of snowmelt and infiltration, soil moisture evaporation and freezing for various types of landuse/landcover (LULC). However, the range of variation of some parameters of the model is quite wide. The model does not calibrate the parameters of HRUs, but the parameters of soil and LULC types for the entire river basin, for example, horizontal and vertical hydraulic conductivity of soil types, snowmelt and evaporation coefficients of LULC types, etc. Each HRU has its own set of soil and LULC types which define the parameters of the model. It is important to emphasize some of the features of manual calibration of the ECOMAG model. First, the values ​​of the key parameters of the land surface (elevation, soil and LULC) are the initial values ​​for calibration, and the task is to find the optimal parameters near these initial values. Secondly, the calibration is organized in order to preserve the relationship between the values ​​of a certain spatially distributed parameter of the soil or LULC type in the catchment. Thus, as a result of calibration, a set of spatially distributed parameters for the entire basin is determined using combination of soil and LULC types.

This information has been added to the Materials and Methods section (Line 128-154 and 241-258).

Motovilov, Y.; Gottschalk, L.; Engeland, K.; Rodhe, A. Validation of a distributed hydrological model against spatial observations. Agricultural and Forest Meteorology 1999, 98-9, 257-277, doi:10.1016/S0168-1923(99)00102-1.

Motovilov, Y. Hydrological Simulation of River Basins at Different Spatial Scales: 1. Generalization and Averaging Algorithms. Water Resources 2016, 43, 429-437, doi:10.1134/S0097807816030118.

Line 118 provide the source links

The source has been added for HWSD and GLCC (Line 205 and 209)

Line 119 please provide the more details such as resolution and date of collection of the image.

GLCC resolution is 1 km. Imagery dates from April 1992 through March 1993. (Line 212-213)

Figure 2 – it should be soil map , b) landuse/landcover map of

Figure 2 updated: Soil map (a) and LULC map (b) of the Iya River basin according to HWSD and GLCC (Line 218)

Line 129-130 It is recommended to show the timeseries plot of temperature and precipitation

Figure 3 has been added (Line 229)

Figure 3. Interannual dynamics of air temperature and precipitation in the mountainous and lowland parts of the Iya River basin over the period 1970–2019

Table 1: It’s a good way to represent hydrometerological datasets however I suggest to author to give convert Table 1 into graph.

The author believes that this information is better read in table form (Table 2) (Line 227). Figure 3 has been added.

Line 141-142 Two indicators are not sufficient to justify the results therefore I would suggest authors to add some more particular volume error, rmse etc.

RSR (ratio of the root mean square error to the standard deviation of measured data) calculation results were added for runoff values similar to KGE and PBIAS (Line 261, 17, Table 3, Table 5).

Line 160-161 What is the basis of this range? Here I would suggest to go through the Moriasi et al., 2007 for model evaluation guidelines.

Moriasi, D. N., Arnold, J. G., Van Liew, M. W., Bingner, R. L., Harmel, R. D., & Veith, T. L. (2007). Model evaluation guidelines for systematic quantification of accuracy in watershed simulations. Transactions of the ASABE, 50(3), 885-900.

The source (Moriasi et al, 2007) has been added (Line 280).

It is not clear from this paragraph which type of calibration have author used in their study for the simulation of model discharge.

A description of the calibration has been added to the Materials and methods section (Line 241-258)

Line 165-166 sentence need restructuring

The sentence updated: A detailed analysis of the precipitation and streamflow regime during the highest floods in the Iya River basin during the observation period was carried out based on observed data and simulation results. The main factors of flood generation were identified according to the methodology in the Amur and Lena River basins [17,31]. (Line 282-285)

Line 168-187 Enlist all the data sources and the citations.

The source has been added for ISIMIP, ERA reanalysis, Climate Research Unit, Global Precipitation Climatology Center (Line 303, 307, 309, 311)

Line 173 is there any specific reason to use this circulation model

According to (Kalugin, 2018), the use of the HadGEM2-ES output data for the river basins of Eastern Siberia and the Far East makes it possible to most effectively (within the ISIMIP project) reproduce meteorological parameters in comparison with the meteorological station data.

Kalugin, A. Variations of the Present-Day Annual and Seasonal Runoff in the Far East and Siberia with the Use of Regional Hydrological and Global Climate Models. Water Resources 2018, 45, S102-S111, doi:10.1134/S0097807818050366.

**Results and Discussion**

Provide a table showing efficiencies for calibration and validation and with the calibrated model parameters.

Tables with the performance criteria of the model, as well as the list of calibration parameters have been added (Line 345, 363)

Table 3. Model performance of the Iya River at the Tulun gauge over the period 1970–2019.

Calibration

Verification

daily discharge

annual peak discharge

hazardous high flow

KGE

PBIAS, %

RSR

R

PBIAS, %

RSR

KGE

PBIAS, %

RSR

0.84

8

0.51

0.97

–3

0.30

0.91

–1

0.41

Table 4. Calibration parameters of the ECOMAG model for the Iya River basin

Parameter

Dimension

Value

Coefficient for horizontal hydraulic conductivity of the topsoil layer

dimensionless

10

Coefficient for vertical hydraulic conductivity of soil type

dimensionless

15

Evaporation coefficient of LULC type

dimensionless

0.35

Baseflow of HRUs

mm day-1

0.11

Precipitation gradient

mm 100 m-1

4

Air temperature gradient

°С 100 m-1

–0.6

Air temperature for transformation of precipitation phase

°С

0.3

Snowmelt air temperature

°С

0.0

Snowmelt intensity for LULC types

mm °С day-1

0.28

From the results section I feel author have provided the hydrograph however I would recommend few things to make the results section more comprehensible.

Include flow duration curves for discharge for calibration and validation period similar to the findings of the obtained in Srivastava et al., 2018. They can add the description from this study. Also author can use the exceedance probability distribution to showcase the frequency distribution.

Srivastava, A., Sahoo, B., Raghuwanshi, N. S., & Chatterjee, C. (2018). Modelling the dynamics of evapotranspiration using Variable Infiltration Capacity model and regionally calibrated Hargreaves approach. Irrig Sci 36, 289–300 (2018). https://doi.org/10.1007/s00271-018-0583-y

Long-term daily flow duration curve (FDC) of the Iya River at Tulun gauge over the period 1970–2019 was plotted as an additional performance of the runoff generation model (Figure 4). The value of RSR for this FDC was 0.28. (Line 346-348)

Figure 4. Long-term daily FDC of the Iya River at Tulun gauge over the period 1970–2019

Also it would be great if authors can show the water balance components obtained from the model for the basin to insights how the model works for each processes.

To represent the performance of the water balance in the hydrological model, a graph of the interannual dynamics of the observed and simulated runoff coefficient of the Iya River at Tulun gauge over the period 1970–2019 is plotted (Figure 5). (Line 351-354)

Figure 5. Interannual observed and simulated runoff coefficient of the Iya River at Tulun gauge over the period 1970–2019

In the hydrographs please add the precipitation on the secondary axis in Figure 3.

Figure 6 (3) updated (Line 412)

**Conclusions**

Authors should mention in the conclusion where the models applied need to be improved or any suggestions for future research in terms of models' performance improvement and application

Information on the prospects of using the presented model for solving various scientific and practical problems has been added to the conclusions. (Line 667-672)

“Given the extreme importance of research in recurring floods for this region and the interest at the state level, the hydrological model presented in this article can be used not only as a method for studying the floods that have occurred, but also for short-term forecasting of future floods based on forecast data from mesoscale weather models, or calculations of possible long-term changes in both the climate and changes in the land surface parameters (for example, deforestation due to felling or fires).”

Author Response

Response to Reviewer 2 Comments

The author is very grateful to the reviewer for effective comments and suggestions.

MS: WATER-1162041

OVERVIEW

This study presents an analysis of floods in the Iya River basin located in the Irkutsk Region (Eastern Siberia). The total catchment area is 18100 km2 (up to the town of Tulun is 14500 km2). The semi-distributed process-based Ecological Model for Applied Geophysics (ECOMAG) was applied for the simulation of catastrophic floods over the observation period and to forecast the impact of future climate changes on the characteristics of the high flows. The analysis showed that floods were caused by a series of extreme precipitations over several days above the mean monthly precipitation on waterlogged soil. The maximum-recorded daily peak discharge was formed at the end of June 2019 under the conditions of the highest averaged-basin pre-flood coefficient of water-saturated soil and 2.5 mean monthly precipitation during three days. According to the calculations of the hydrological model based on future climate scenarios, one can expect not so much a change in the annual peak discharge or precipitation, but rather a change in the runoff during the period of exceeding the critical water level and its duration.  

The MS is of interest to the WATER readership and is written in an acceptable manner, though the English language sounds strange in some parts. I have appreciated the computational effort and I didn’t find evident drawbacks. However, novelties and advantages in knowledge this manuscript would provide remain unclear. It appears this study is no longer an application of available methodologies rather than a presentation of new approaches/results. Moreover, the description of methods as well as the presentation of results is rather shallow and some issues would deserve more discussion.

In conclusion, this MS could be worthy of publication in Water journal. Its quality is already quite acceptable. However, I would suggest the following specific comments in the hope that they might improve the quality of this paper. A re-review is recommended. 

SPECIFIC COMMENTS [R# major and r# minor concerns]

[-] Title. This manuscript is mainly a Case Study and I have appreciated that in the title this aspect has been highlighted.

[R1] Introduction. I would emphasize, at the end of this section, the novelties and advances in knowledge (for instance from the methodological point of view) this study would provide in comparison to similar literature studies. 

Response: Comment on the overview and R1.

The author agrees that the material presented in this article does not significantly deepen knowledge in the field of hydrology in the global sense. As indicated by the reviewer, the direction of local research for a specific catchment is emphasized in the title of the article. However, the author would like to clarify that despite the regular flood events on the Iya River, there was no physically-based runoff generation model for this river basin until 2020. Then, given the importance and challenges of the 2019 maximum flood, attempts were made to adapt runoff generation models for this catchment using either hydrometeorological observation timeseries for a short period of the 20th century (e.g. Fedorova et al, 2020) or models focused only on the 2019 flood (e.g. Shalikovsky et al, 2019).

Accordingly, the development of a process-based hydrological model with distributed parameters (the main method of this study) using a continuous long-term period of hydrometeorological observations from 1970 to 2019, including recent years, and its verification by the accuracy of calculating high runoff, will make it possible to carried out a comprehensive analysis of the formation conditions of the highest floods in within a unified methodological approach. This is one aspect of the purpose of this study. Another aspect is that the applied method of process-based hydrological modeling makes it possible to obtain the results of the influence of future climate changes on the physical transformation of the high flow of the Iya River, which may differ significantly from the historical period. This is also a new and, as the author hopes, useful result.

Given the extreme importance of research in recurring floods for this region and the interest at the state level, the hydrological model presented in this article can be used not only as a method for studying the floods that have occurred, but also for short-term forecasting of future floods based on forecast data from mesoscale weather models, or calculations of possible long-term changes in both the climate and changes in the land surface parameters (for example, deforestation due to felling or fires).

Thus, the range of applied use of the results presented in this article is very diverse, while it is of local importance. The author considered it important to emphasize this by adding relevant information to the introduction (Line 99-108 in the attached file water-1162041_editing.pdf) and conclusions (Line 667-672).

Fedorova, A.; Makarieva, O.; Nesterova, N.; Shikhov, A.; Vinogradova, T. Modelling maximum discharge of the  catastrophic flood at the Iya River (Irkutsk  region, Russia) in 2019  E3S Web of Conferences 2020, 163, 01004, doi:10.1051/e3sconf/202016301004.

Shalikovsky, A.; Lepikhin, A.; Tiunov, A.; Kurganovich, K.; Morozov, М. The 2019 Floods in Irkutsk Region. Water Sector of Russia 2019, 6, 48-65, doi:10.35567/1999-4508-2019-6-4.

[R2] Materials and Methods. I find this section somewhat shallow. The Author should provide more information. Namely:

(i) Distribute hydrological models imply the setting of many parameters related, for instance, to the rainfall interception, snowmelt, transpiration and evaporation, infiltration, percolation and capillary rise, groundwater flow, overland runoff and streamflow processes [e.g. Xingguo Mo et al. 2006, HSJ]. However, the Author did not discuss/describe these parameters and the other possible ones;

The author fully agrees that a detailed description of the processes of the applied hydrological model and its calibration should be added.

In a model schematization of a river basin, its surface is divided into subbasins (hydrological response units – HRUs) based on a digital elevation model and the structure of the river network. Modeling of hydrological processes at each HRU is performed for four levels: the topsoil layer, horizon of caliche, groundwater and prechannel flow. During the cold season, snow cover is added. Prechannel flow is formed after filling the depressions of the land surface due to the formation of excess water, which does not infiltrate and flows down the slopes of the catchment to the river network. As a result of infiltration in the soil, water moves in the aeration zone along a slope to the channel network or transforms into a groundwater zone. In the model, the subsurface and groundwater flow is described according to the Darcy equation, and the prechannel and stream flow is described by the kinematic wave equation. The total porosity in the soil aeration zone is divided into capillary and non-capillary zones. It is assumed that when soil moisture is less than the field capacity, all soil water is in the capillary zone, and when soil moisture is greater than the field moisture capacity, it is in the non-capillary zone. In conditions of high soil moisture, the actual evaporation is equal to the potential, and then it decreases linearly to zero as the soil moisture decreases to wilting point. Potential evaporation is estimated according to the Dalton method.

The snowmelt rate is calculated using the degree-day method. The phase transformation of precipitation depends on the air temperature. The evaporation of solid and liquid phases of snow is estimated using data on the air humidity deficit. It is assumed that vertical temperature profiles in snow, frozen and thawed soil differ insignificantly from linear ones, and moisture migration to the soil freezing front is insignificant. Under these conditions, the dynamics of the depth of freezing and thawing of the soil can be described by a system of differential equations. Infiltration of rain and melt water into frozen soil is calculated taking into account the effect of ice content in frozen soil on the hydraulic conductivity of the soil. A more detailed mathematical description of the flow generation processes in the ECOMAG model is presented by its author in (Motovilov et al, 1999; Motovilov, 2016).

Most of the parameters of the ECOMAG model for the Iya River basin are set a priori based on Harmonized World Soil Database (HWSD) and Global Land Cover Characterization (GLCC). Such soil parameters as bulk density, porosity, field capacity, wilting point, hydraulic conductivity are determined for each soil type using pedotransfer functions and soil grain-size distribution data. The model uses the coefficients of snowmelt and infiltration, soil moisture evaporation and freezing for various types of landuse/landcover (LULC). However, the range of variation of some parameters of the model is quite wide. The model does not calibrate the parameters of HRUs, but the parameters of soil and LULC types for the entire river basin, for example, horizontal and vertical hydraulic conductivity of soil types, snowmelt and evaporation coefficients of LULC types, etc. Each HRU has its own set of soil and LULC types which define the parameters of the model. It is important to emphasize some of the features of manual calibration of the ECOMAG model. First, the values ​​of the key parameters of the land surface (elevation, soil and LULC) are the initial values ​​for calibration, and the task is to find the optimal parameters near these initial values. Secondly, the calibration is organized in order to preserve the relationship between the values ​​of a certain spatially distributed parameter of the soil or LULC type in the catchment. Thus, as a result of calibration, a set of spatially distributed parameters for the entire basin is determined using combination of soil and LULC types.

This information has been added to the Materials and Methods section (Line 128-154 and 241-258).

Motovilov, Y.; Gottschalk, L.; Engeland, K.; Rodhe, A. Validation of a distributed hydrological model against spatial observations. Agricultural and Forest Meteorology 1999, 98-9, 257-277, doi:10.1016/S0168-1923(99)00102-1.

Motovilov, Y. Hydrological Simulation of River Basins at Different Spatial Scales: 1. Generalization and Averaging Algorithms. Water Resources 2016, 43, 429-437, doi:10.1134/S0097807816030118.

(ii) Moreover, did the Author assess the sensitivity of these parameters to the model performance?

Given the study of the high runoff of the Iya River over the summer period, the hydrological model is most sensitive to the calibration parameters associated with horizontal and vertical hydraulic conductivity of soil types, as well as evaporation of LULC types. This information has been added to the Results and Discussion section (Line 359-362).

(iii) Could the Author discuss the most frequent generating mechanisms (e.g. rainfall, snowmelt, ice-dams) of floods? 

During the observation period 1936–2019, the highest annual peak discharge of the Iya River is characteristic over the summer period: 12 events in June, 49 events in July, 23 events in August. The average long-term date of annual peak discharge is July 20. Since the air temperature in the catchment becomes positive in the third decade of April, annual peak discharge of the Iya River is associated with liquid precipitation and not with snowmelt or possible ice-dams in spring. This information has been added to the Section 3.2 (Line 370-375).

[R3] Results and discussion.

(i) At lines 292 and 293 it reads “The main natural factors for the formation of floods on the Iya River were extreme liquid precipitation and the previous waterlogging due to a number of heavy rains”. However, how did the Author simulate the infiltration mechanism? How did the Author validate these results?

A description of the infiltration process is now presented in the materials and methods section, as well as in the response to comment R2. Unfortunately, it is difficult to verify the accuracy of the soil moisture calculation in comparison with the observational data due to their absence for the Iya River basin. To verify the model by soil moisture, detailed measurement data are required, since this is an extremely variable in the catchment area, and the calculation of the averaged-basin soil moisture based on the interpolation of measurement data at several points will lead to large error. Taking into account the use of a calibrated process-based model, which quite effectively reproduces the high flow of the Iya River, it was assumed that the simulated soil moisture values ​​can be used to analyze the formation conditions of floods. Such a comment has been added to the Results and Discussion (Line 466-472).

(ii) In sub-section 3.3 [Impact of climate change on the high flow of the Iya River] did the Author consider the impact of climate change on floods generated by snowmelt and/or ice-dams?

The response to comment R2 explains in detail that over 84 years of observations on the Iya River, the maximum flow is caused by liquid precipitation over the summer period. Therefore, continuous calculations to assess the impact of the future climate on the change in the high runoff of the Iya River were also carried out only over the summer period, when snowmelt and/or ice-dams could not affect the flood generation.

All the above issues need a discussion in the manuscript. Moreover, the following minor/small changes are needed:

[r1] Abstract. The acronym RCP should be spelled out. In other terms, it reads “various RCP-scenarios”, but I would write “various Representative Concentration Pathways (RCP-scenarios)”.

The acronym RCP moved from section 3.3 to abstract. (Line 26)

[r2] Keywords. The keywords “climate change” and “high flow” appear too broad and ineffective in identifying the crucial issues of this manuscript. I would substitute them for more specific/ suitable ones.

Some keywords have been replaced: change in maximum flow; the ECOMAG model; ISIMIP (Line 33)

[r3] At line 483. It reads “we recommend organizing”, but I would write “the Author recommends organizing”.

The author agrees, this replacement has been made (Line 664).

Round 2

Reviewer 1 Report

The authors have addressed all previous concerns expressed by the reviewers and in the process have improved the work, confirmed the validity of their findings and gained confidence in their results and conclusions. I would like to congratulate the authors for a interesting and well executed work and I recommend this manuscript for publication in its current form.

Reviewer 2 Report

The Authors addressed all my concerns carefully. The manuscript is better discussed now and presented in a well-argued manner. Its style should be improved, but this can be done at the proofreading stage. In conclusion, I would recommend accepting this manuscript as it is.